# A synthetic biology approach to probing nucleosome symmetry

Yuichi Ichikawa[1†], Caitlin F Connelly[2†], Alon Appleboim[3,4], Thomas CR Miller[5‡], Hadas Jacobi[3,4], Nebiyu A Abshiru[6], Hsin-Jung Chou[2], Yuanyuan Chen[1], Upasna Sharma[2], Yupeng Zheng[6], Paul M Thomas[6], Hsuiyi V Chen[2], Vineeta Bajaj[1], Christoph W Müller[5], Neil L Kelleher[6], Nir Friedman[3,4], Daniel NA Bolon[2], Oliver J Rando[2]*, Paul D Kaufman[1]*

[1]Department of Molecular, Cell and Cancer Biology, University of Massachusetts Medical School, Worcester, United States; [2]Department of Biochemistry and Molecular Pharmacology, University of Massachusetts Medical School, Worcester, United States; [3]School of Computer Science and Engineering, The Hebrew University of Jerusalem, Jerusalem, Israel; [4]The Alexander Silberman Institute of Life Sciences, The Hebrew University of Jerusalem, Jerusalem, Israel; [5]Structural and Computational Biology Unit, European Molecular Biology Laboratory, Heidelberg, Germany; [6]National Resource for Translational and Developmental Proteomics, Northwestern University, Evanston, United States

*For correspondence:
Oliver.Rando@umassmed.edu
(OJR);
paul.kaufman1@umassmed.edu
(PDK)

[†]These authors contributed
equally to this work

Present address: [‡]Molecular
Machines Laboratory, The
Francis Crick Institute, London,
United Kingdom

Competing interests: The
authors declare that no
competing interests exist.

Reviewing editor: Jerry L
Workman, Stowers Institute for
Medical Research, United States

**Abstract** The repeating subunit of chromatin, the nucleosome, includes two copies of each of the four core histones, and several recent studies have reported that asymmetrically-modified nucleosomes occur at regulatory elements in vivo. To probe the mechanisms by which histone modifications are read out, we designed an obligate pair of H3 heterodimers, termed H3X and H3Y, which we extensively validated genetically and biochemically. Comparing the effects of asymmetric histone tail point mutants with those of symmetric double mutants revealed that a single methylated H3K36 per nucleosome was sufficient to silence cryptic transcription in vivo. We also demonstrate the utility of this system for analysis of histone modification crosstalk, using mass spectrometry to separately identify modifications on each H3 molecule within asymmetric nucleosomes. The ability to generate asymmetric nucleosomes in vivo and in vitro provides a powerful and generalizable tool to probe the mechanisms by which H3 tails are read out by effector proteins in the cell.
DOI: https://doi.org/10.7554/eLife.28836.001

## Introduction

Packaging of eukaryotic genomes into chromatin has widespread consequences for genomic function, as nucleosomes generally impede the access of proteins to the underlying DNA (*Hughes and Rando, 2014*). In addition, the histone proteins are subject to a huge variety of covalent modifications, which affect genomic function in ways ranging from direct alterations in the biophysical nature of the chromatin fiber to recruitment or activation of various regulatory proteins.

The histone octamer is comprised of two copies of each of the four histones H2A, H2B, H3, and H4, organized with two H2A/H2B dimers associated with a central core tetramer of histone H3 and H4 (*Kornberg and Lorch, 1999*). The symmetry inherent to the histone octamer has long been appreciated, and it has been suggested that chromatin function may be regulated at the level of histone modification symmetry – that K4me3/K4me0 and K4me3/K4me3 nucleosomes might serve distinct functions in vivo, for example. Indeed, increasing evidence supports the existence of

asymmetrically-modified nucleosomes in vivo. In mammals, immunopurification and mass spectrometry of nucleosomes marked with H3K27me3 revealed a mixture of symmetrically-modified nucleosomes as well as asymmetric nucleosomes modified on only one H3 tail (*Voigt et al., 2012*). More recent single molecule studies of H3K4me3/K27me3-marked 'bivalent' nucleosomes reveal that the vast majority of these nucleosomes are asymmetrically modified, with each H3 tail bearing one of the two methylation marks, rather than both marks co-occurring on the same H3 molecule (*Shema et al., 2016*). In budding yeast, ChIP-Exo and chemical mapping studies identified a subset of nucleosomes that exhibit asymmetric contacts with the underlying genomic DNA (*Rhee et al., 2014*; *Ramachandran et al., 2015*), suggested to reflect histone octamers associated with an asymmetrically-localized DNA bulge.

While evidence therefore increasingly supports the hypothesis that nucleosomes across the genome – particularly at regulatory regions – can be modified asymmetrically, at present it is largely unknown whether nucleosome symmetry affects genomic functions such as transcription. For instance, do any chromatin regulators bind specifically to singly-modified nucleosomes, or to doubly-modified nucleosomes? A handful of studies in vitro have suggested that chromatin regulators can distinguish singly-modified from doubly-modified nucleosomes. For instance, Li *et al* assembled and purified asymmetric nucleosomes lacking a single H3 tail in vitro, subsequently showing that the histone acetyltransferase Gcn5 exhibited cooperative binding to two histone tails (*Li and Shogren-Knaak, 2008*; *Li and Shogren-Knaak, 2009*). Conversely, asymmetrically H3K4-methylated nucleosomes were, like unmodified nucleosomes, efficiently methylated at K27 by the PRC2 complex, whereas this methylation was inhibited by the presence of symmetric H3K4me3 marks (*Voigt et al., 2012*). Although these in vitro studies generate hypotheses for how chromatin effectors might be affected by nucleosome symmetry in the cell, the inherent symmetry of the natural H3-H3 interface has made in vivo tests of these hypotheses impossible until now.

Here, we develop a novel system to enable analysis of the functional consequences of nucleosome symmetry in vivo. Using rational design, followed by in vivo optimization and selection, we developed a pair of H3 proteins – H3X and H3Y – that form heterodimers in vivo, yet cannot homodimerize. We validated this system genetically and biochemically in budding yeast, confirming that the vast majority of histone octamers in the cell contain a heterodimeric H3X/H3Y at their core. Furthermore, we confirm that the engineered specificities are recapitulated in vitro during biochemical reconstitution of histone octamers with recombinant subunits. We then used this system to probe several mechanistic aspects of nucleosome function in vivo, finding that a single copy of H3K36 is sufficient for stress resistance, and to silence cryptic transcription, in vivo. In addition, we show that disruption of H3S10 affects H3K9 acetylation only on the same tail, demonstrating the utility of this approach for interrogating histone 'crosstalk'. This system thus represents a powerful approach to mechanistically probe the role of histone stoichiometry and symmetry in a variety of aspects of chromatin biology.

## Results

### Design and validation of obligate heterodimers

The histone octamer is built on a central core tetramer of histone H3 and H4, assembled around a central H3-H3 homodimeric interface (*Luger et al., 1997*). In order to develop a system to analyze the role of nucleosome symmetry in chromatin regulation in vivo, we set out to design a pair of H3 molecules – 'H3X' and 'H3Y' – which cannot form X/X or Y/Y homodimers, but will efficiently form H3X/H3Y heterodimers (*Figure 1A*). We initially used protein design to develop six sets of H3 mutants that are predicted to have much higher affinity with a paired mutant protein than with wild-type H3. Each engineered variant has 'bumps and holes' that are also predicted to reduce the potential for homodimerization. We focused our efforts on the hydrophobic cluster formed by amino acids L109, A110, L126, and L130 (*Figure 1B*) in the nucleosome structure (pdb 1KX3). At these positions, sequence and conformational space were enumerated using a rotamer description of side chain geometry and a molecular mechanics force-field (*Mayo et al., 1990*) to generate a pairwise energy matrix. We used a previously described algorithm (*Bolon et al., 2005*) to identify sequences that preferentially energetically partition as heterodimers.

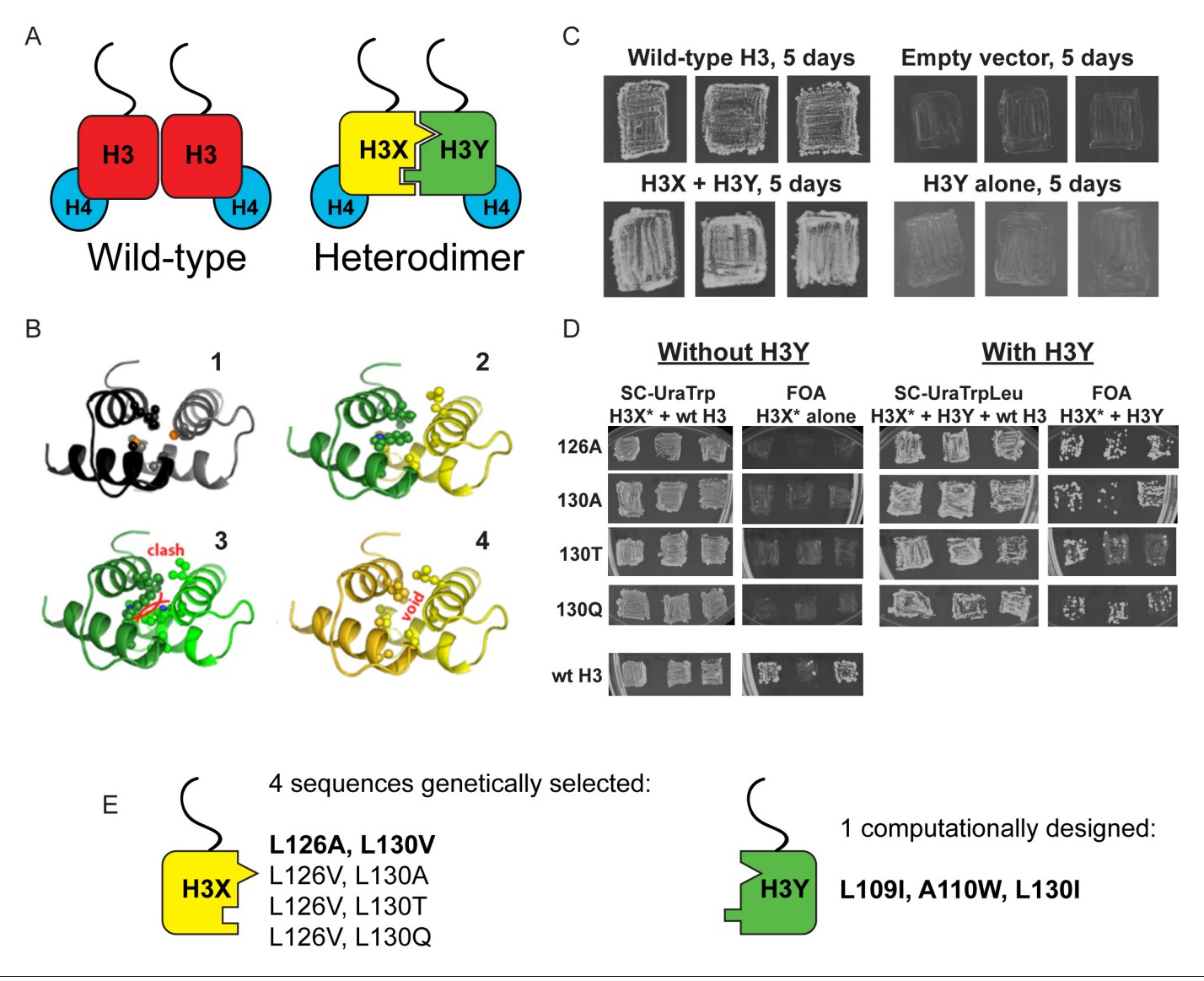

**Figure 1.** Design and testing of asymmetric nucleosomes. (A) Schematic of asymmetric histone H3 design. Left: Wild-type H3/H4 tetramers are symmetrical, with H3-H3 interactions serving as the dimerization interface. The H3 N-terminus, site of many modifiable residues, is indicated protruding from the globular domain. Right: Obligate heterodimer H3s are comprised of distinct 'X' and 'Y' interaction partners which are altered to prevent either H3 from homodimerizing. (B) Computational models of the designed histone H3 heterodimer. (1) The four-helix bundle comprising the H3-H3 C-terminal dimerization interface (pdb 1KX3). Leucine and alanine residues at the hydrophobic interface are indicated. (2) The designed asymmetric mutations pack efficiently as a heterodimer, with partners colored in green (H3Y) and yellow (H3X). Indicated residues at the H3-H3 interface were engineered to form a bump-hole interface to increase interaction affinity. (3-4) Altered H3s are designed not to homodimerize due to van der Waals clashes (3) or voids (4) in the hydrophobic core. (C–D) Genetic analysis of heterodimeric H3X/H3Y pairs. C)H3Y alone cannot support growth. Images show growth of yeast carrying wild-type H3 on a *URA3*-marked plasmid as well as either empty plasmid, wild-type H3, H3Y alone, or both H3X and H3Y. Three independent transformants for each strain were grown on 5-FOA to select against the *URA3* shuffle plasmid. (D) H3X alone cannot support growth. Left panels: growth of three independent transformants of various H3X strains (evolved second-generation mutations indicated at left) in the absence of H3Y. Bottom: positive control with a *TRP1*-marked wild-type histone plasmid. Right panels: growth of indicated H3X strains in the presence of H3Y. Synthetic complete (SC) plates were grown 3 days, FOA plates 9 days. No growth on FOA was observed for any of the four evolved H3X alleles in the absence of H3Y, even after extended incubation. (E) Sequences of final H3X-H3Y molecules. All four H3X variants have been validated genetically, while two variants – 126A/130V and 126V/130A – have been validated biochemically (see *Figure 2* and not shown). 126A is used for all functional analyses.

DOI: https://doi.org/10.7554/eLife.28836.002

The following figure supplement is available for figure 1:

**Figure supplement 1.** Optimization of the H3X/H3Y design.

*Figure 1 continued on next page*

*Figure 1 continued*

DOI: https://doi.org/10.7554/eLife.28836.003

We then carried out genetic tests of six initial designs in vivo, reasoning that budding yeast carrying both H3X and H3Y should be viable, but yeast carrying H3X alone or H3Y alone would be inviable due to the failure of these H3 molecules to homodimerize. In our initial tests, one of the original six designs (H3X = L126V, L130V; H3Y = L109I, A110W, L130I) exhibited most of the expected growth behaviors. However, although yeast carrying H3Y alone exhibited no growth even after 21 days of culture (*Figure 1C* and not shown), H3X-only strains did grow even in the absence of H3Y, albeit poorly – colonies were only observed after nine or more days of growth of these strains (*Figure 1—figure supplement 1A*), in contrast to the abundant colonies observed for H3X/H3Y strains after just two or three days. As homodimerization of H3X even at low levels could complicate interpretation of this system, we carried out a round of in vivo selection to optimize the H3X interface (*Figure 1—figure supplement 1B–C*). We randomized each of the H3/H3 interface residues on the backbone of the initial H3X gene, generating libraries of yeast strains bearing these mutant 'H3X*' molecules paired with our previously-selected H3Y design. Strains that grew robustly with H3X*/H3Y pairs were then re-tested for growth in the absence of the H3Y. In this manner, we identified four final versions of H3X (named 126A, 130A, 130T, and 130Q for the additional substitutions on the original H3X backbone) that reproducibly conferred robust growth in the presence of H3Y but were completely inviable in its absence (*Figure 1D–E*). We obtained multiple isolates encoding each of these amino acid substitutions (13, 5, 5, and 5, respectively), suggesting that we had adequately surveyed the possible codons. Consistent with our observation that these residues are sensitive points for H3/H3 dimeric interactions, L126A and L130A mutations cause lethality when present on symmetric H3 molecules (*Dai et al., 2008*).

Although these genetic tests suggest that H3X or H3Y alone do not homodimerize efficiently enough to support growth, the possibility remains that in the context of cells expressing both H3X and H3Y there could nonetheless exist a substantial fraction of inappropriate homodimeric interactions, which would complicate interpretation of downstream results. To quantitatively determine the extent of X/X or Y/Y homodimerization in vivo, a biotin-tagged (*Beckett et al., 1999*) H3X or Y variant was coexpressed in combination with two other H3s: the same variant with a different epitope tag, and the designed partner (*Figure 2*). To analyze H3-H3 interactions within individual nucleosomes and avoid contamination from adjacent nucleosomes (as could arise from undigested dinucleosomes), we treated extracts from these cells with dimethyl suberimidate (DMS), a crosslinking agent that produces well-characterized crosslinks within histone octamers (*Kornberg and Thomas, 1974*; *Thomas, 1989*). Crosslinked samples were then digested with micrococcal nuclease to generate a soluble population of mononucleosomes. This was followed by biotin-based affinity purification in the presence of high salt (2M NaCl) to remove DNA and adjacent nucleosomes, leaving only crosslinked protein complexes that were subsequently analyzed by Western blotting. The DMS crosslinking efficiency of the X-Y heterodimeric pairs was nearly identical to that observed in cells expressing tagged H3 variants with a wild-type, homodimeric interface (*Figure 2—figure supplement 1*).

Using this assay, we analyzed the extent of homodimerization of a lead H3X candidate using strains expressing Biotin-H3X, V5-H3X, and Myc-H3Y (*Figure 2A–C*). After streptavidin purification of biotin-H3X from DMS-crosslinked yeast, the species migrating at 31kD (red arrow, *Figure 2B*) is comprised of the captured biotin-H3X crosslinked to the other H3 from the same nucleosome. Myc-H3Y molecules were abundantly detected in the crosslinked 31kD species (*Figure 2B*, *Figure 2—figure supplement 2*), consistent with the expected heterodimerization with H3X. In contrast, barely detectable levels of V5-H3X – which would result from H3X-H3X homodimerization – were observed in the bound 31kD species (*Figure 2B*, quantitated in *Figure 2C*). In a reciprocal experiment, we assayed H3Y homodimerization, again finding efficient heterodimerization in vivo along with barely detectable H3Y homodimerization (*Figure 2D–F*, *Figure 2—figure supplement 2*).

In addition to its utility for genetic analysis of nucleosome symmetry, the ability to generate asymmetric nucleosomes in vitro would also provide a valuable resource for biochemical and structural studies. We therefore purified recombinant human histones H2A, H2B, and H4, as well as

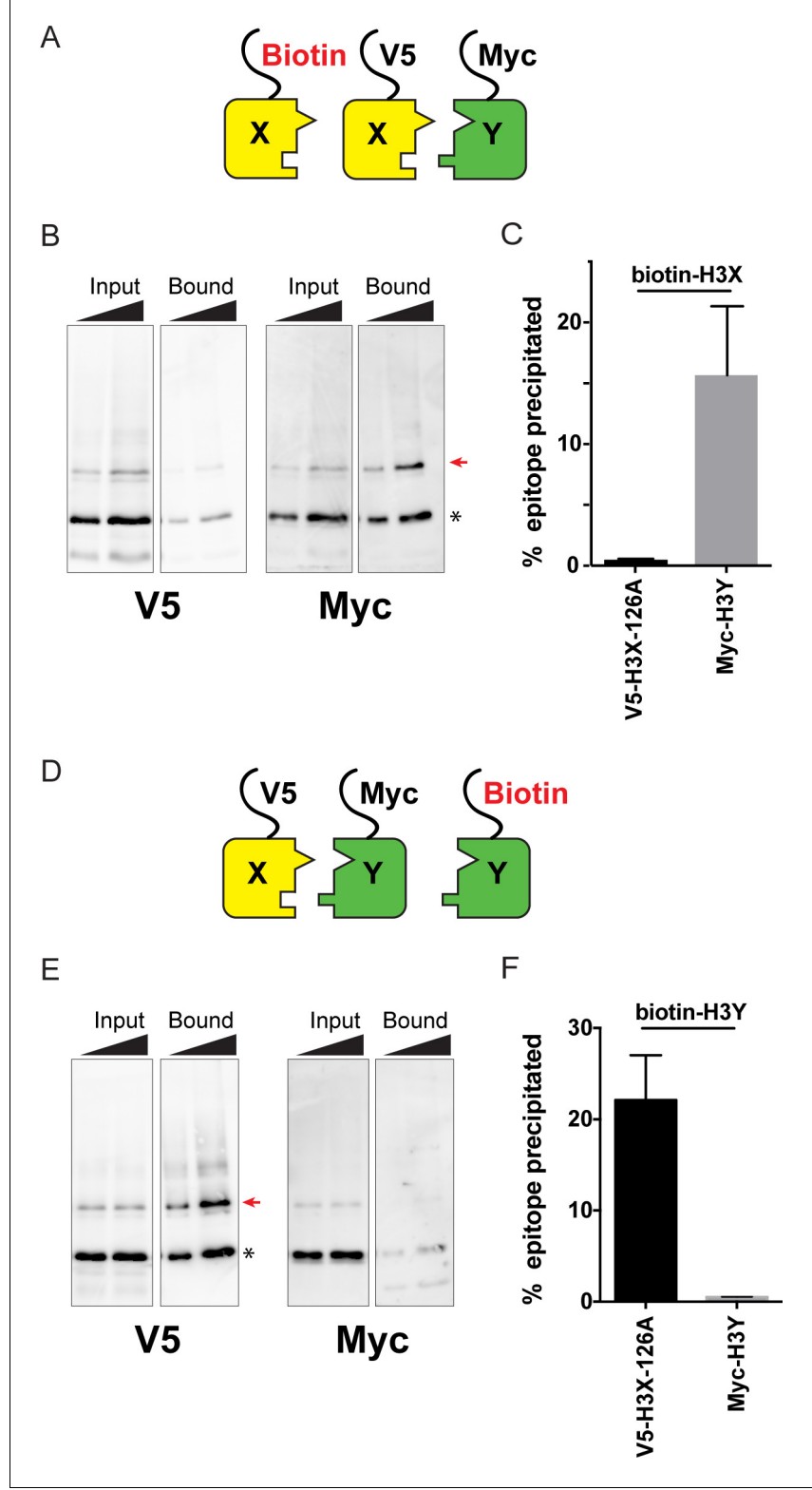

**Figure 2.** Biochemical analysis of asymmetric nucleosome formation in vivo. (**A**) Schematic for in vivo biochemical analysis of H3X dimerization. Yeast strains expressed biotin-tagged H3X along with tagged H3X and H3Y, as indicated. (**B**) Immunoblot analysis of biotin-H3X partners. For each of the indicated antibodies (Myc or V5, the same blot was probed sequentially), the left two lanes show total DMS-crosslinked, MNase-digested chromatin (Input), and right lanes show streptavidin-precipitated biotinylated-H3 (Bound). Arrow indicates the expected size

*Figure 2 continued on next page*

*Figure 2 continued*

(31 kD) of crosslinked H3-H3 dimer species – note that biotin-H3 molecules crosslinked to H4 would not be recognized by the anti-epitope antibodies used for Western blot analysis. Asterisk indicates monomeric H3 molecules – presence of V5-tagged H3X (left panel) at this position in the bound fraction reflects bead contamination by uncrosslinked H3 molecules. Full gel, including unbound material and DMS control, is shown in *Figure 2—figure supplement 2*. (C) Quantitation of the amounts of the crosslinked 31kD H3-H3 dimers precipitated with streptavidin-agarose for second-generation H3X variant 126A. The mean and standard deviation for three independent replicate experiments are shown. (D–F) As in (A–C), but for biotin-tagged H3Y. As above, the homodimer signal (Myc, right panel) at 31kD in the bound material was almost undetectable, while V5-tagged H3X/H3Y heterodimers were readily detected. The mean and standard deviation for three independent replicate experiments are quantitated in panel (F).

DOI: https://doi.org/10.7554/eLife.28836.004

The following figure supplements are available for figure 2:

**Figure supplement 1.** DMS crosslinking efficiency.
DOI: https://doi.org/10.7554/eLife.28836.005

**Figure supplement 2.** Western blot analysis of asymmetric nucleosome formation in vivo.
DOI: https://doi.org/10.7554/eLife.28836.006

---

recombinant H3X and H3Y variants engineered onto the human H3 histone backbone. Gel filtration of in vitro assembled octamers revealed robust assembly of histone octamers assembled using wild-type H3, or assembled around the H3X/H3Y heterodimer pair. In contrast, attempts to refold histone octamers using H3X only or H3Y only resulted in high molecular weight aggregates with no functional histone octamer (*Figure 3*, *Figure 3—figure supplement 1A–B*). Moreover, H3X/H3Y-based octamers were readily reconstituted into nucleosomes when assembled with DNA, whereas the material obtained from H3X alone or H3Y alone did not assemble into nucleosomes.

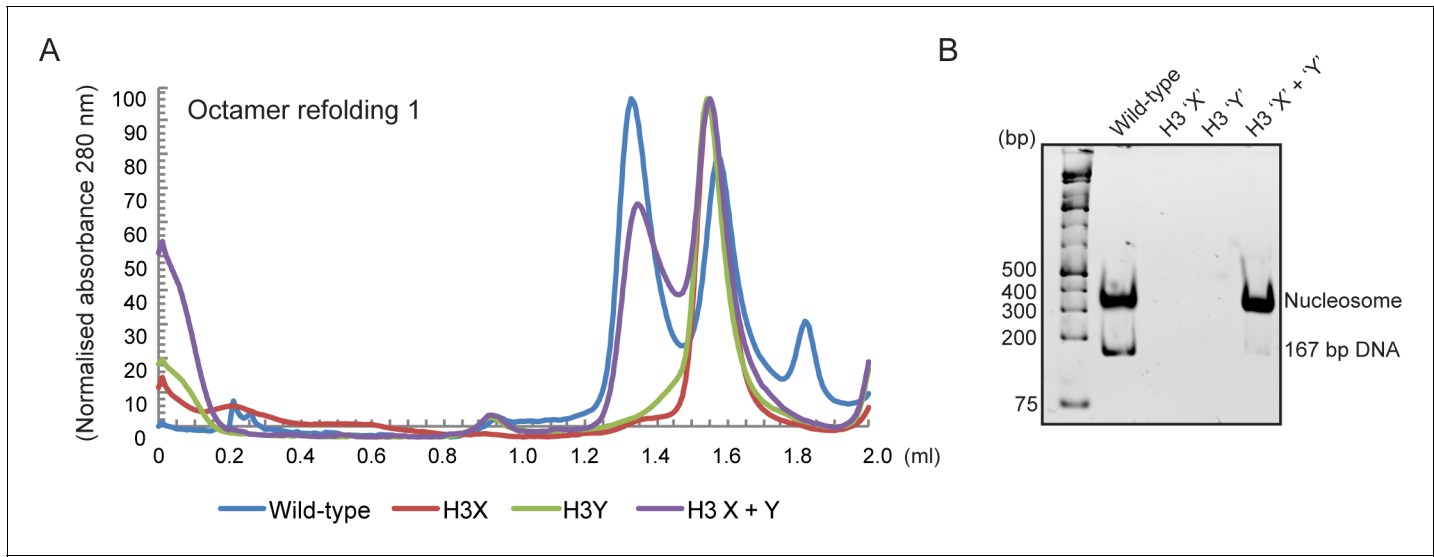

**Figure 3.** Human histone H3 mutants 'X' and 'Y' form obligate heterodimers in vitro. (A) Size exclusion (Superdex 200 Increase 3.2/300) gel filtration profiles showing the purification of octamer refolding reactions containing wild-type and heterodimer 'X' and 'Y' histone H3. Octamer refolding in the presence of both 'X' and 'Y' H3 yields histone octamers with a characteristic elution profile, similar to refolding reactions containing wild-type H3, while reconstitutions with only H3X or only H3Y form aggregates. (B) Native PAGE analysis of nucleosome reconstitution experiments with purified histones from (A). H3X/Y octamer reconstitutions readily form nucleosomes when assembled onto DNA, while the aggregates formed in octamer refolding reactions containing either 'X' or 'Y' alone do not form histone octamers and therefore cannot form nucleosomes in vitro.

DOI: https://doi.org/10.7554/eLife.28836.007

The following figure supplement is available for figure 3:

**Figure supplement 1.** Generation of asymmetric X/Y nucleosomes for in vitro studies.
DOI: https://doi.org/10.7554/eLife.28836.008

In addition to the ability to generate asymmetric nucleosomes from recombinant proteins in vitro, chromatin purified from mutant yeast strains has also proven to be a powerful tool for in vitro chromatin studies (see, eg, (*Altaf et al., 2007*)). We therefore affinity-purified chromatin from strains expressing an asymmetric H3X/H3Y pair in which one H3 carries a biotin tag, readily recovering the core histones regardless of whether the biotin tag was present on the X or Y variant (*Figure 3—figure supplement 1C*). The engineered H3X/H3Y interface thus enables the convenient generation of asymmetric nucleosomes for biophysical and biochemical studies of nucleosomal symmetry in vitro.

Taken together, our genetic and biochemical analyses validate our designed obligate heterodimers, indicating that the great majority (average of ~97% as measured in *Figure 2C,F*) if not all of the nucleosomes in vivo are indeed heterodimeric. We have therefore established a unique system for manipulation of nucleosomal symmetry in vivo and in vitro. As with all studies based on expression of mutant histones, the mutant nucleosomes will be located throughout the yeast genome. Nonetheless, such studies have provided great insights into chromatin function over decades of study (*Dai et al., 2008*; *Johnson et al., 1990*), distinguishing between genes whose expression requires a given histone residue and those genes that are unaffected despite the loss of the residue (and its modification, if present).

## Genetic analysis of interactions between histone tail mutations

The ability to generate asymmetric nucleosomes in vivo enabled us to quantitatively characterize interactions between mutations on the two histone tails using genetic epistasis. To this end, we generated 'trios' of histone point mutants – asymmetric single mutations H3Xwt/H3Y**mut** and H3X**mut**/H3Ywt, and the double mutation H3X**mut**/H3Y**mut**. Importantly, although strains expressing the heterodimeric X/Y H3s exhibit reduced growth in YPD (doubling times of ~240 min at 30°C compared to ~140 min for the parent strain, *Figure 4—figure supplement 1A*) and a modest temperature-sensitive phenotype (robust growth at 34 C but failure to grow at 37 C – see below), any effects of the mutant X/Y interface on chromatin regulation and cellular function are internally controlled for in all studies by comparing all yeast carrying single or double histone mutations to the 'pseudo-wild type' strain (H3Xwt/H3Ywt).

To broadly survey the landscape of inter-tail epistasis between histone mutations, we generated a set of 12 mutant trios, focusing on relatively well-characterized sites of post-translational covalent modifications. Mutant effects on organismal phenotypes are often context-dependent, and stress conditions can provide a sensitized environment for detection of chromatin-related phenotypes (*Weiner et al., 2012*). We first assayed the relative fitness of these strains in both low (0.2 M) and high (0.8 M) KCl-containing media (*Figure 4A*), as the osmotic stress response is one of the most commonly studied transcriptional responses in budding yeast (*Capaldi et al., 2008*; *Gasch et al., 2000*). As expected, growth rates for strains carrying a single mutation on H3X were generally well-correlated with those for H3Y mutants, although a small subset of mutations exhibited different behavior on different H3 backbones (see, eg, H3K37Q in 0.8M KCl).

Consistent with the often-subtle phenotypes resulting from histone point mutations, the majority of mutations examined conferred no significant growth defects under either growth condition, or subtle growth effects such as the enhanced growth observed in all three H3K18Q strains growing in high salt. In contrast, we noted significantly diminished growth, particularly in high salt, for yeast carrying double H3K14R, H3K14Q, or H3K36Q mutations (*Figure 4B*, *Figure 4—figure supplement 1B*). Interestingly, the behaviors of the corresponding asymmetric single mutants were distinct: for H3K14R, single mutants already exhibited a partial growth defect that was exacerbated in the double mutant, while for H3K36Q the single mutant strains had no measurable deficit compared to the pseudo-wild type. These growth phenotypes correspond roughly to additive and recessive epistasis, respectively. Importantly, the stress sensitivity observed for these mutants was not specific to osmotic stress. Consistent with prior studies demonstrating that yeast lacking H3K36 methylation are hypersensitive to DNA damaging agents (*Jha and Strahl, 2014*), we find that the K36Q double mutants – but neither of the single K36Q mutants – displayed increased sensitivity to both the DNA alkylating agent methyl methanesulfonate (MMS) and the strand break-inducing agent phleomycin (*Figure 4C*). Moreover, these mutants were also temperature-sensitive, as symmetric double mutants exhibited reduced growth at 34 C, while the asymmetric mutants exhibited similar or only slightly reduced growth compared to the pseudo-wild type under these conditions (*Figure 4D*, *Figure 4—figure supplement 2*). We conclude that K36Q mutations confer sensitivity to osmotic, temperature

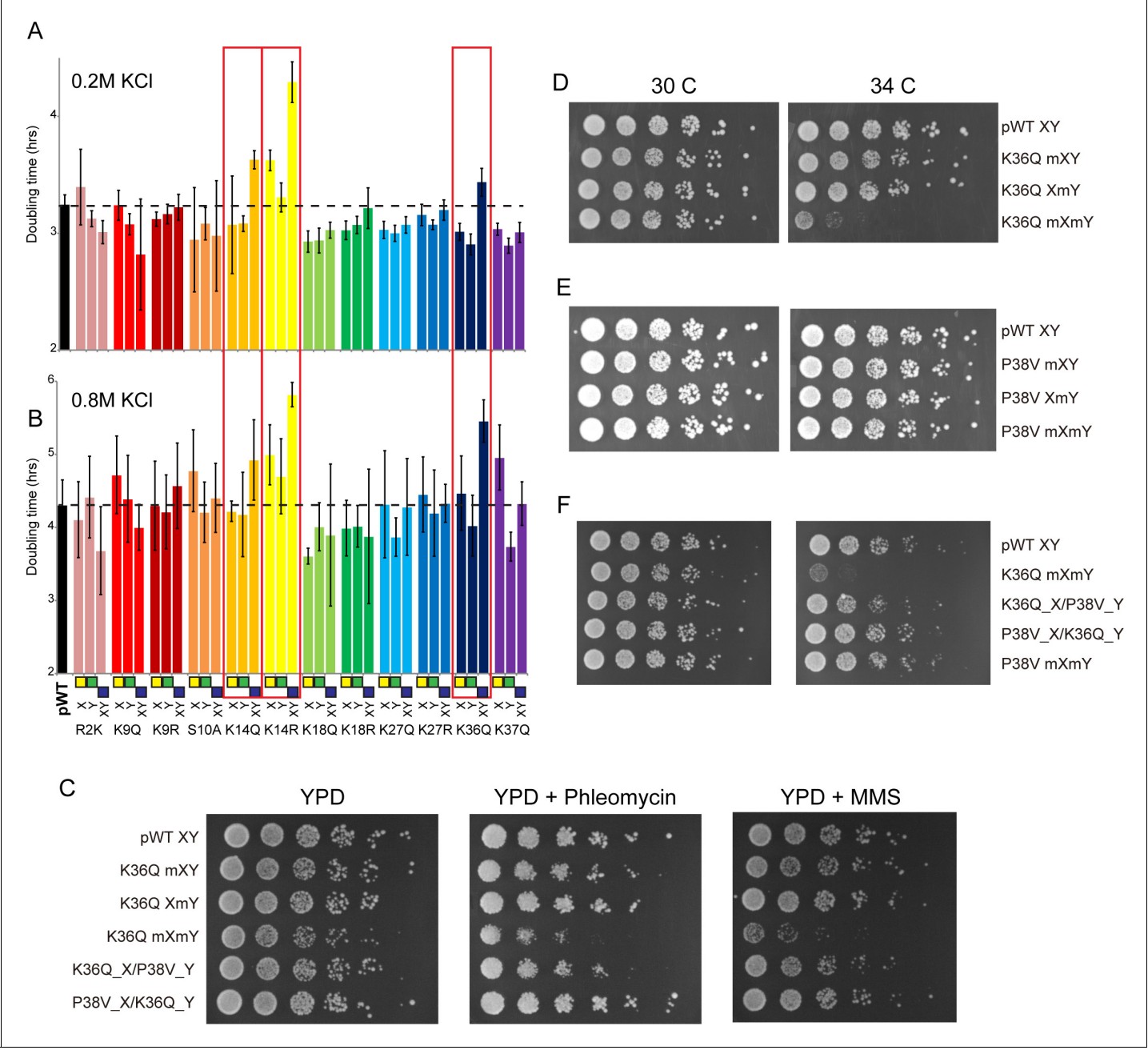

**Figure 4.** Genetic epistasis between the H3 tails. (A–B) We constructed 12 sets of point mutations in histone H3, with each mutant cycle including H3Xmut/H3Ywt, H3Xwt/H3Ymut, and H3Xmut/H3Ymut – for each mutant trio, X/Y/double mutations are ordered from left to right, as indicated with green/yellow/blue boxes in Panel (B). For all 12 trios, as well as the matched 'pseudo-wt' (labeled as **pWT**) carrying H3Xwt/H3Ywt (leftmost bar and dashed line), we measured doubling times in at least six replicate cultures in low (A) and high (B) KCl. Boxes emphasize H3K14R and H3K36Q trios, as detailed in text. Note that growth rates here were measured robotically in a 96-well format, so doubling times are slower than those typically measured in well-aerated cultures (*Figure 4—figure supplement 1A*). (C) Effects of H3K36Q mutants on DNA damage sensitivity. Serial dilutions of the indicated strains were plated on YPD, YPD + 10 ng/ml phleomycin or YPD + 0.025% MMS, as indicated. pWT XY (PKY4704), K36Q mXY (PKY4829), K36Q XmY (PKY4831), K36Q mXmY (PKY4834), K36Q_X/P38V_Y (PKY5138), and P38V_X/K36Q_Y (PKY5140). (D–F) Serial dilution growth assay for the indicated strains incubated at 30 C or 34 C, as indicated. Note that yeast carrying pseudo-wild type H3X/H3Y nucleosomes grow poorly at 37C, so we assayed temperature sensitivity at 34 C. Mutant set in (D) shows temperature-sensitivity is specific to symmetric H3K36Q mutant. Mutant set in (E) shows that complete loss of H3K36me3 in the symmetric H3P38V mutant (where H3K36me3 is replaced with H3K36me2) is compatible with rapid growth at high temperatures. Mutant set in (F) shows that a single dimethylated K36 residue per nucleosome is compatible with rapid growth at high temperatures. Strains analyzed were: pWT XY (PKY4704), R2K mXY (PKY4749), R2K XmY (PKY4751), R2K mXmY (PKY4753), K9Q mXY (PKY4714), K9Q XmY (PKY4715), K9Q mXmY (PKY4706), K9R mXY (PKY4773), K9R XmY (PKY4775), K9R mXmY (PKY4777), S10A mXY (PKY4716), S10A XmY (PKY4717), S10A mXmY

*Figure 4 continued on next page*

*Figure 4 continued*

(PKY4707), K14Q mXY (PKY4789), K14Q XmY (PKY4791), K14Q mXmY (PKY4793), K14R mXY (PKY4781), K14R XmY (PKY4783), K14R mXmY (PKY4786), K18Q mXY (PKY4805), K18Q XmY (PKY4807), K18Q mXmY (PKY4809), K18R mXY (PKY4797), K18R XmY (PKY4799), K18R mXmY (PKY4801), K27Q mXY (PKY4822), K27Q XmY (PKY4823), K27Q mXmY (PKY4825), K27R mXY (PKY4813), K27R XmY (PKY4815), K27R mXmY (PKY4817), K36Q mXY (PKY4829), K36Q XmY (PKY4831), K36Q mXmY (PKY4834), K37Q mXY (PKY4837), K37Q XmY (PKY4839), K37Q mXmY (PKY4841), P38V mXY (PKY5033), P38V XmY (PKY5035), P38V mXmY (PKY5037), K36Q mX/P38V mY (PKY5138), and P38V mX/K36Q mY (PKY5140). pWT XY, pseudo WT; mXY, mutation on X; XmY, mutation on Y; and mXmY, mutation on both X and Y.

DOI: https://doi.org/10.7554/eLife.28836.009

The following figure supplements are available for figure 4:

**Figure supplement 1.** Significant growth defects for the H3K36Q double mutant, relative to either single mutant.

DOI: https://doi.org/10.7554/eLife.28836.010

**Figure supplement 2.** Temperature sensitivity of various histone mutant strains.

DOI: https://doi.org/10.7554/eLife.28836.011

**Figure supplement 3.** H3P38V mutations block trimethylation but not dimethylation of H3K36.

DOI: https://doi.org/10.7554/eLife.28836.012

and DNA damage stresses, but only when symmetrically present on nucleosomes. These data support the biochemical data indicating that we can separately manipulate single H3 molecules within nucleosomes in vivo, and provide an illustration of the types of biological regulation that can be discovered using these tools.

## A single H3K36 suffices to silence cryptic internal promoters

Our growth data that a single copy of H3K36 per nucleosome is sufficient to support growth under stress conditions. What is the mechanistic basis for this function? In principle, mutation of this lysine could (1) affect biophysical properties of the nucleosome by eliminating a positive charge, (2) disrupt a binding site for regulatory protein that binds to the unmodified tail, or (3) disrupt (or mimic) a modification-dependent regulatory event (*Rando and Winston, 2012*). H3K36 is methylated by Set2 (*Strahl et al., 2002*), with the resulting modified residues serving to activate the RPD3S histone deacetylase complex via binding of the Eaf3 chromodomain-containing subunit (*Carrozza et al., 2005*; *Keogh et al., 2005*; *Joshi and Struhl, 2005*; *Govind et al., 2010*; *Drouin et al., 2010*). As the major biological role served by H3K36 methylation is silencing of cryptic internal promoters via activation of the RPD3S complex (*Carrozza et al., 2005*), and given that RPD3S can bind to nucleosomes with a single H3K36me3 modification (*Huh et al., 2012*), we sought to test whether a single methylated H3K36 per nucleosome would be sufficient to support robust silencing of such cryptic transcripts in vivo.

We first explored this idea with genetic tests, comparing the growth phenotypes of a P38V mutant series to K36Q mutants. The P38V mutants enable di- and tri-methylation of H3K36 to be distinguished – the wild-type proline residue at this position undergoes regulated isomerization, and P38V mutations that eliminate the possibility of proline isomerization block trimethylation (*Nelson et al., 2006*) but not dimethylation (*Youdell et al., 2008*) of the nearby K36 residue, a finding that we confirm here (*Figure 4—figure supplement 3*). Importantly, cells expressing only H3P38V display none of the cryptic transcription phenotypes observed in *set2*, *eaf3*, or H3K36Q mutants ([*Youdell et al., 2008*] and see below), indicating that H3K36me2 is sufficient for the proper activation of the Rpd3S complex with regards to repression of cryptic transcription. Consistent with this, we find that double P38V mutations, on either a wild-type H3 backbone or in the context of the XY system, did not cause the temperature sensitivity observed in K36Q mutant cells (*Figure 4E*, *Figure 4—figure supplement 2*). We therefore hypothesized that a single H3 N-terminal tail bearing K36 di- or trimethylation is sufficient to repress cryptic transcription.

To determine whether the single methylated K36 must be trimethylated or whether dimethylation is sufficient, we generated strains carrying the K36Q and P38V mutations in a trans conformation, that is, on opposite H3 N-termini (*Figure 4F*). Consistent with previous analyses of the effects of P38V (*Nelson et al., 2006*; *Youdell et al., 2008*), immunoblotting confirms that these strains have one H3 tail (K36Q) that lacks K36 methylation entirely, and one tail (P38V) with a dimethylated K36 residue (*Figure 4—figure supplement 3*). We made strains with both possible configurations: H3XK36Q + H3YP38V and H3XP38V + H3YK36Q. Notably, neither of these K36Q/P38V trans strains

are sensitive to growth at 34C (*Figure 4F*) or DNA damaging agents (*Figure 4C*). Thus, they pheno-typically resemble the P38V double mutants and the single K36Q mutants, rather than the temperature-sensitive K36Q double mutants. These data indicate that a single K36me2 is sufficient to promote stress tolerance in budding yeast.

To determine whether the growth data reflect underlying defects in control of cryptic transcription, we carried out Northern blots against a canonical target of this pathway – *STE11* – which is host to a well-characterized cryptic promoter that is silenced in a K36- and RPD3S-dependent manner (*Carrozza et al., 2005*; *Keogh et al., 2005*; *Joshi and Struhl, 2005*). First, we confirmed the expected increase in cryptic transcription in *eaf3Δ* mutants, and in yeast carrying symmetric H3K36Q mutations on the wild-type H3 backbone (*Figure 5A–B*, lanes 1–4, 10). In addition, we confirm that H3K36me2 was sufficient to silence the *STE11* cryptic promoter, as H3P38V mutants (which lack H3K36me3 but exhibit normal H3K36me2 levels) did not cause significantly increased cryptic transcription (lane 5).

Turning next to the question of whether one H3K36 is sufficient for silencing, we analyzed *STE11* RNAs in single and double H3K36Q mutants in the H3X/Y backbone (*Figure 5*, lanes 6–9). Consistent with the growth phenotypes observed in *Figure 4*, we find that the pseudo-wildtype H3X/Y cells and the cells with a single K36Q mutated tail display similar, low levels of cryptic transcripts (lanes 6–8). In contrast, the cells with two K36Q mutated tails display greatly elevated cryptic transcript levels (lane 9). As expected, neither single nor double P38V mutant tails affected cryptic transcription in our strain background (lanes 14–16). Finally, P38V/K36Q trans strains (lanes 17–18) displayed modest derepression of cryptic transcription, more similar to that observed in single K36Q mutant cells (lanes 7–8) than the high level observed in K36Q double mutants (lane 9). We confirmed the generality of these findings using another well-studied model for cryptic transcription, the *FLO8* locus (*Kaplan et al., 2003*), where derepression of cryptic transcription was again observed specifically in the K36Q double mutant, but not in either K36Q single mutant (*Figure 5—figure supplement 1*). We conclude that a single K36-di-methylated H3 tail per nucleosome is sufficient to restrain cryptic transcription in vivo.

## Interrogating histone crosstalk in cis and trans

In addition to our ability to uncover effects of asymmetric mutant nucleosomes on transcriptional regulation, our system also provides a unique opportunity to determine whether histone 'crosstalk' – the ability of one histone modification to alter the level of another modification via effects on modifying enzyme recruitment or activity (*Suganuma and Workman, 2008*) – occurs in cis (on the same histone), or in trans. We generated biotin-tagged H3 molecules either carrying a point mutation of interest in cis, or carrying an otherwise wild-type sequence on the biotin-tagged H3, paired with a point mutation in trans (*Figure 6A*, cartoon). Biotin-tagged H3 molecules were then purified under stringent, partially denaturing conditions to facilitate the separation of H3X and H3Y, and the biotin-tagged H3 was subject to mass spectrometry to identify histone modifications present. Analysis of peptides derived from the engineered H3X/H3Y interface confirmed our ability to efficiently purify the biotin-tagged H3 molecule away from the untagged H3 molecule from each nucleosome, with >93% purity observed for informative peptides analyzed (*Figure 6A*).

As cases of histone crosstalk that occur between different histones (e.g. effects of H2B ubiquitylation on H3 methylation levels) cannot be interrogated using our system, we focused on H3S10, whose phosphorylation affects histone H3 lysine acetylation (*Lo et al., 2000*; *Liokatis et al., 2016*). We found that the fraction of H3 molecules acetylated at K9 was lower in yeast strains carrying H3S10A symmetrically, either on a wild-type histone backbone or in the H3X/H3Y system. This decrease was observed by mass spec, and confirmed by quantitative Western blot (*Figure 6B and C*, left panel). Turning next to analysis of asymmetric mutant nucleosomes, mass spectrometric analyses showed that H3K9 acetylation levels were significantly decreased on the same histone tail carrying the S10A mutation, but were not affected by the presence of H3S10A in trans (*Figure 6B*). Western blotting for H3K9ac in yeast strains where an epitope tag allows H3X and H3Y isoforms to be separated by gel electrophoresis confirmed our mass spec-based analysis of histone acetylation, with H3S10A affecting adjacent H3K9 acetylation on the same tail, but having no effect on modification of a paired wild-type H3 tail (*Figure 6C–F*).

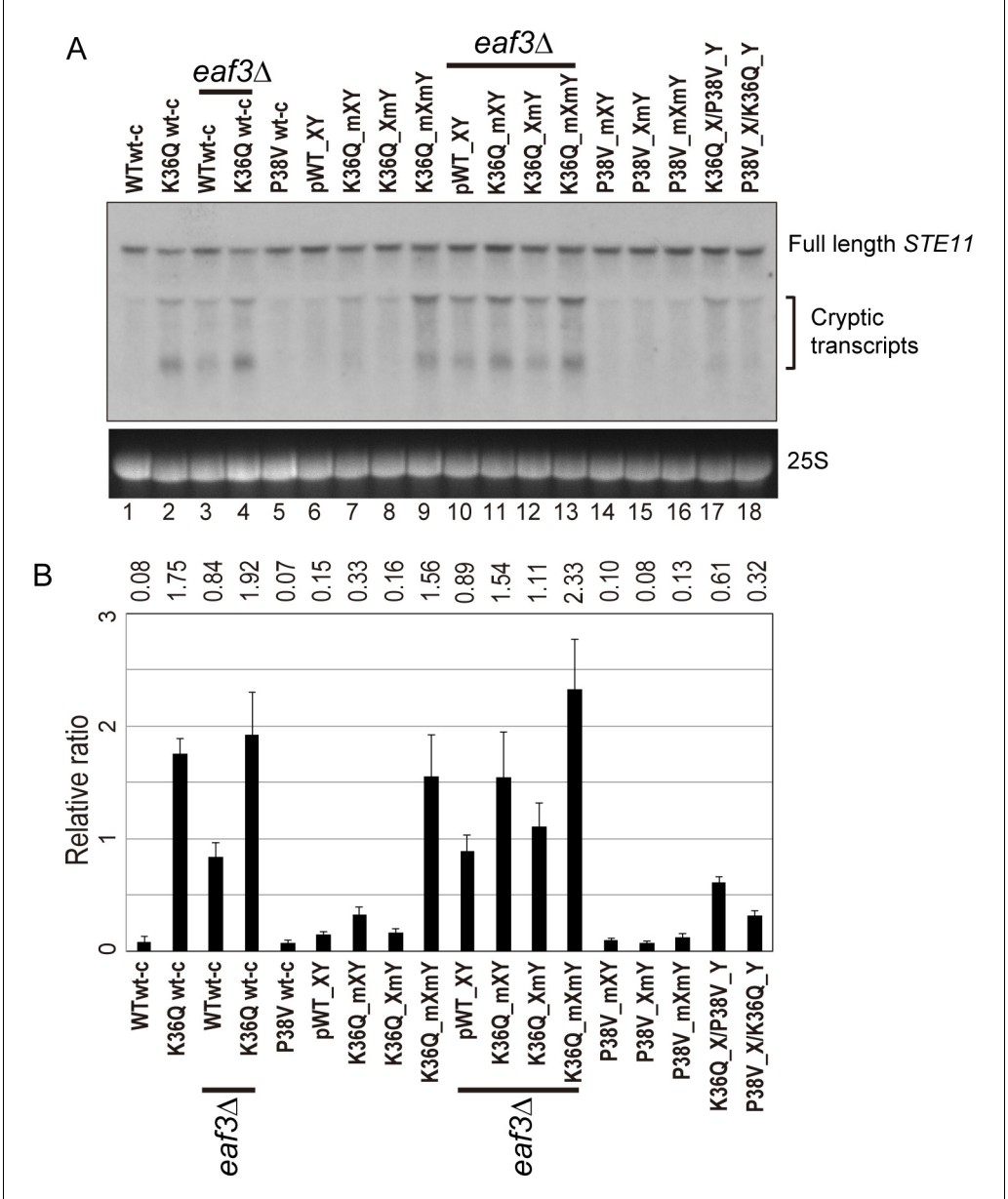

**Figure 5.** Double H3K36Q mutants fail to repress cryptic promoters. (A) Northern blot for *STE11* for RNA isolated from the indicated strains, with 25S Northern shown as a loading control. Top band, which migrates at 2.2 kb, represents the full-length *STE11* transcript, while the two lower molecular-weight bands correspond to previously-characterized sense transcripts initiating within the *STE11* coding region from cryptic promoters (**Kaplan et al., 2003**). (B) Quantitation of cryptic transcript levels for *STE11* in the various strains indicated (n = 3). Levels of cryptic transcripts were normalized to full-length. Average and std. dev. of triplicate measures of these normalized values are graphed on the y-axes. Strains analyzed were: WT wt-c (PKY4701), K36Q wt-c (PKY4827), *eaf3Δ* WT wt-c (PKY5077), *eaf3Δ* K36Q wt-c (PKY5079), P38V wt-c (PKY5031), pWT XY (PKY4704), K36Q mXY (PKY4829), K36Q XmY (PKY4831), K36Q mXmY (PKY4834), *eaf3Δ* pWT XY (PKY5081), *eaf3Δ* K36Q mXY (PKY5083), *eaf3Δ* K36Q XmY (PKY5086) and *eaf3Δ* K36Q mXmY (PKY5087), P38V mXY (PKY5033), P38V XmY (PKY5035), P38V mXmY (PKY5037), K36Q mX/P38V mY (PKY5138), and P38V mX/K36Q mY (PKY5140). pWT_XY, pseudo wild-type; mXY, mutation on X; XmY, mutation on Y; and mXmY, mutation on both X and Y.

DOI: https://doi.org/10.7554/eLife.28836.013

The following figure supplement is available for figure 5:

**Figure supplement 1.** Effects of H3K36Q mutants on cryptic transcription.

DOI: https://doi.org/10.7554/eLife.28836.014

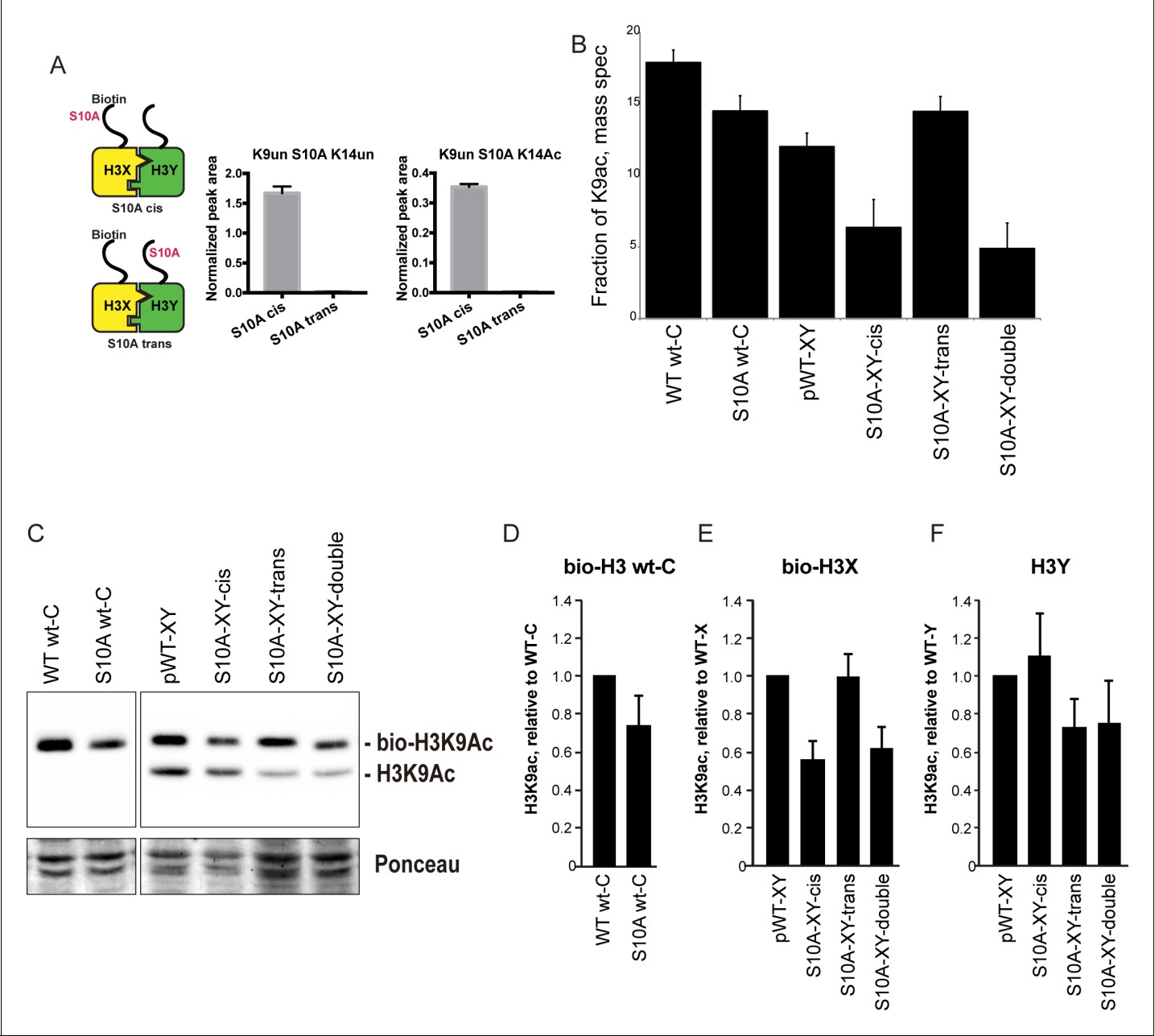

**Figure 6.** H3S10A affects histone crosstalk in cis. (**A**) Mass spectrometric analysis of H3X/H3Y heterodimers expressing one biotin-labeled subunit, with a S10A point mutation either located on the same H3 molecule (in cis), or in trans. Left: Schematic of asymmetric nucleosomes. Right: Robust avidin-affinity purification of biotin-labeled H3 molecules. Mass spec quantitation of peptides (described on top of the graphs) from the strains with genotypes indicated on the x-axes. For each peptide, raw peak area scores were normalized by dividing by the peak areas of an internal control, the unmodified H3 peptide containing K42. Average and std. dev. of triplicate measures of these normalized values are graphed on the y-axes. As expected, S10A-containing peptides were only detected when present in cis on the biotin-labeled histone, but not when expressed in trans on the heterodimeric partner. (**B**) H3K9 acetylation levels are diminished in cis in H3S10A mutants. Data are shown for mass spec analysis of the indicated mutant strains. Left two bars show data for wild-type and double H3S10A mutants on the background of a wild-type H3-H3 interface. Right four panels show H3K9Ac levels on an avidin-purified biotin-tagged H3 molecule, showing data for H3X acetylation in the context of the pseudo-wild type H3X/Y background, H3X acetylation on the same (cis) or opposite tail from an H3S10A mutation, and H3X acetylation in a double H3XS10A/H3YS10A mutant. (**C**) Western blots confirming effects of symmetric S10A on K9 acetylation both on the WT H3-H3 background, and on the X-Y background, as indicated. For the right panel, the epitope tag present on H3X allows separate probing of the H3X and H3Y molecules. Total protein on blots was visualized by Ponceau S staining. Note that although the S10A mutation in principle might affect anti-H3K9ac antibody (Abcam ab10812) binding, these results are highly concordant with the mass spectrometry measurements in panel (**B**), where the ability to detect lysine acetylation (based on peptide mass) is unaffected by the difference between S10 and S10A. (**D–F**) Quantitation of blots from panel (**C**). K9Ac signals were normalized to total protein detected by

*Figure 6 continued on next page*

*Figure 6 continued*

Ponceau S. Strains analyzed were: WT wt-C (PKY4610), S10A wt-C (PKY5003), pWT-XY (PKY4983), S10A-XY-cis (PKY5005), S10A-XY trans (PKY4986) and S10A-XY double (PKY5042).
DOI: https://doi.org/10.7554/eLife.28836.015

## Discussion

Here, we demonstrate a novel system enabling genetically-encoded symmetry breaking for the histone octamer. Using protein engineering and directed evolution, we have designed and validated a pair of H3 variants – H3X and H3Y – that efficiently heterodimerize with one another in vivo, but which form extremely low levels (<3%) of homodimers. The ability to generate nucleosomes bearing either single mutations or symmetric double mutations to key regulatory residues will allow detailed mechanistic analyses of the roles for histone modifications in various aspects of chromosome biology. Here, we demonstrate two of many potential uses of this system, interrogating the role for H3K36 in control of cryptic transcription, and investigating 'crosstalk' between histone modifications.

### Interpreting epistasis between the histone tails

To genetically interrogate interactions between the histone tails, we analyzed phenotypes in yeast strains carrying one of four nucleosomes: H3Xwt/H3Ywt – 'pseudo-wild-type' – and H3X**mut**/H3Ywt, H3Xwt/H3Y**mut**, and H3X**mut**/H3Y**mut**. In principle, we could expect to see a variety of classes of interactions between the mutations, including transgressive epistasis (not observed in this study), and a range from dominant epistasis, in which the single mutants exhibit the same phenotype observed in the double mutant, to recessive epistasis, in which a given phenotype is only observed in the double. For quantitative traits, these correspond to alleviating and aggravating epistasis, respectively (*Segrè et al., 2005*).

Measured in the context of obligate heterodimers, such genetic interactions can provide insights into the mechanistic basis for readout of histone modifications. As a simple example, a protein that binds cooperatively to both tails of a given nucleosome should give rise to dominant epistasis between the histone mutations that affect its binding (depending of course on the concentration of the factor in question and its Kd for a single tail), as loss of a single residue abrogates the cooperative binding. Conversely, if each histone tail recruits a binding protein independently of the other, then loss of a single tail could have a range of quantitative effects on local gene expression, depending on whether or not the binder is typically recruited in excess for some function. Downstream of initial binding interactions, many readouts of histone modification function, such as mRNA abundance, are several mechanistic steps removed from direct binding, and thus more complex models can readily be envisioned that would exhibit a range of distinct epistasis behaviors depending on the input/output relationships for the various intervening steps.

Here, we surveyed intra-tail epistasis for 12 sets of H3 point mutants, simply using growth rate as a phenotype. Even with this coarse resolution, we identified inter-tail interactions including dominant epistasis (for the enhanced low-salt growth seen in H3K18Q mutants, for example) to relatively unusual cases of recessive epistasis (for H3K36Q effects on growth in stress conditions). For followup studies we focused on H3K36Q, given that recessive epistasis makes more concrete mechanistic predictions, and given the well-understood role for H3K36 modification in gene regulation.

### A single H3K36me2 per nucleosome is sufficient for stress resistance and repression of cryptic transcripts

Although H3K36 is potentially subject to multiple distinct modification pathways – H3K36 is acetylated at a small fraction of nucleosomes (*Morris et al., 2007*), for example – the majority of phenotypes of H3K36 mutants result from loss of H3K36 methylation-dependent activity of the RPD3S complex (*Carrozza et al., 2005*; *Keogh et al., 2005*; *Joshi and Struhl, 2005*). Our system allowed us to ask, essentially, whether a single RPD3S binding site per nucleosome is sufficient to repress cryptic internal promoters.

Several lines of evidence demonstrate that a single H3K36me2 is sufficient for RPD3S silencing at cryptic promoters. First, we find that H3K36Q significantly affects growth rate – particularly under

stress conditions – only when present in both H3 copies per nucleosome. Next, consistent with a primary function of H3K36 modification being to repress cryptic intragenic promoters, we show that a single wild-type K36 is sufficient to prevent transcription at such promoters. Third, extending prior studies, we find that growth rate and cryptic promoter silencing are both unaffected by double P38V mutants. P38V inhibits trimethylation but not dimethylation of K36 residues (*Youdell et al., 2008*), and nucleosomes with two H3P38V tails are sufficient to prevent cryptic transcription either in natural, symmetric nucleosomes (*Youdell et al., 2008*) or in our asymmetric nucleosomes (*Figure 5*, lane 16). Finally, the K36Q/P38V trans strains indicate that a single dimethylated K36 residue is sufficient to prevent stress sensitivity (*Figure 4F*) and repress cryptic transcripts (*Figure 5*, lanes 17–18). Together, these data indicate that a single K36-dimethylated H3 tail per nucleosome is sufficient to prevent cryptic transcription, and to promote stress resistance.

Notably, the RPD3S complex appears to be particularly well-suited to carrying out its regulatory functions even on chromatin that is less than fully methylated at H3K36. In vitro experiments show that RPD3S is able to bind to nucleosomal templates containing asymmetric H3K36me3 modifications (*Huh et al., 2012*). Further, although RPD3S binds with higher affinity to dinucleosomes than monosomes, it binds with high affinity to templates with a dimethylated nucleosome adjacent to a fully unmethylated one. These data suggest that the methyl-K36/RPD3S system would be buffered against transient reductions in H3K36me2-3 density, for example during DNA repair or replication-mediated loss of modified histones. Our data extend these observations to indicate that the enzyme can function effectively in vivo even if it can only engage a single H3K36me2 or me3 per nucleosome.

## H3S10 mutants affect H3K9 acetylation in cis

As another demonstration of the utility of the H3X/H3Y system, we carried out mass spectrometry analysis of a histone 'crosstalk' pathway. Here, we take advantage of biotin-based purification to stringently separate a biotin-tagged H3X from an untagged H3Y (and vice versa) prior to mass spectrometry analysis. This method allows probing of histone crosstalk pathways in which one modification affects a second modification site. We show that H3S10A mutations decrease acetylation at the neighboring K9 residue, but only on the molecule in cis. This is consistent with a recent in vitro study demonstrating that S10 phosphorylation stimulates H3 tail acetylation by Gcn5 only in cis (*Liokatis et al., 2016*). This example provides a clear example of how asymmetric nucleosomes can be used for detecting modification dependency relationships.

## Potential applications

In addition to the two proof-of-concept analyses described above, we anticipate that the H3X/Y scaffold will provide a powerful tool for many additional studies, and we list a few potential applications here. First, we show here that the H3X/Y system assembles appropriately in vitro (*Figure 3*), providing a convenient method for generation of asymmetric nucleosomes for in vitro studies. This method should prove more generalizable than existing methods that rely on immunopurification (*Voigt et al., 2012*; *Li and Shogren-Knaak, 2008*) or extensive chemical synthesis (*Lechner et al., 2016*). Such asymmetric nucleosomes could be used to generate single-tailed nucleosomes for structural studies of tail-dependent nucleosome-binding proteins, significantly reducing the conformational heterogeneity resulting from the many flexible, unstructured histone tails. Nucleosomes carrying either asymmetric point mutations or asymmetric modified residues (generated using peptide ligation, for example – [*Shogren-Knaak et al., 2003*]) will facilitate detailed biochemical analyses of chromatin regulators, or selection of asymmetry-specific binding partners. Such studies can also be envisioned using asymmetric nucleosomes purified from living cells – thus occupying a wide range of DNA sequences and carrying a multitude of covalent modifications – as affinity-tagged X and Y halves of the heterodimer allow for efficient and rapid purification of chromatin composed of asymmetric nucleosomes (*Figure 3—figure supplement 1C*).

In addition to improving on extant methods to generate asymmetric nucleosomes in vitro, our system uniquely allows genetically-encoded assembly of asymmetric nucleosomes across the genome in living cells. We describe here one example of in vivo analysis of the gene regulatory consequences resulting from single vs. double H3 mutations for H3K36Q, which can of course be extended to additional H3 mutations and to genome-wide or single-cell analyses. Although such

studies are demonstrated here using the budding yeast system, the H3-H3 interface is generally highly conserved throughout evolution, and we expect that this system should function appropriately in other species – indeed, the biochemical reconstitution of asymmetric nucleosomes described in *Figure 3* is carried out using human histone sequences. Although canonical histone genes are often encoded in many copies in large multi-gene genomic clusters and are therefore inconvenient for genetic studies, a *Drosophila* model has been developed in which the histone genes have been relocated to an artificial locus (*McKay et al., 2015*). Along with *S. pombe*, these models should enable analysis of H3K9/HP1 and H3K27/Polycomb repression systems in vivo. In addition, even in other model organisms we expect that low copy number H3 molecules such as H3.3 and CENP-A should prove amenable to analysis using the H3X/Y system.

In any of these systems, the X/Y heterodimer will not just allow analysis of single vs. double mutations of the same residue, but will also enable cis and trans dissection of effects of combinatorial mutations (eg H3XK9R/H3YK14R vs. H3XK9RK14R/H3YWT). Fusing individual H3 molecules to labeling reagents such as BirA or Dam, or introduction of cysteine residues that can be used to create local nicks in the DNA, could potentially also be used to probe stereospecificity – facing from the promoter into a coding region, does a given nucleosome remodeler bind the left or the right H3 tail? – of nucleosome binding proteins. It is also possible to envision pulse-chase methods for investigation of H3/H4 dimer mixing during replication and replication-independent nucleosome replacement.

Together, our data describe a versatile system for investigation of nucleosome symmetry, and demonstrate the utility of in vivo analysis of asymmetric nucleosomes as a mechanistic probe for histone modification function.

## Materials and methods

### Plasmid constructions and mutagenesis

C-terminal mutations of *HHT2*, one of the yeast genes encoding histone H3, were generated according to QuickChange Site-Directed Mutagenesis protocol (Stratagene, La Jolla, CA). Plasmids containing Myc- or Flag-tagged *HHT2* were described previously (*Dion et al., 2007*). The insertion of the V5 or biotin acceptor tags at the 5' end of *HHT2* was generated by PCR. The DNA fragment containing the *E. coli BirA* gene (encoding biotin-protein ligase) (*Beckett et al., 1999*) was cloned into a *HIS3*-marked integration vector downstream of the yeast *GPD* promoter and upstream of *PGK* terminator, and integrated into the *his3* locus of strain PKY4171.

### Yeast strains and growth

Yeast growth and plasmid transformation of yeast cells were done according to standard yeast protocols. For plate assays, strains were grown overnight in 2 ml YPD medium at 30°C, five-fold dilutions (starting from $OD_{600}$ = 0.6) were spotted on YPD-agar plates (with added compounds such as MMS when indicated), and incubated at indicated temperatures for 3–4 days. Analyses of osmotic stress sensitivity (*Figure 4*, *Figure 4—figure supplement 1B*) were performed in robotically-manipulated cultures grown in 96-well plates – the growth rates in these experiments are internally consistent but distinct from growth measurements obtained in well-aerated liquid cultures (*Figure 4—figure supplement 1A*).

Low-copy plasmids containing mutated histone genes were introduced by LiOAc transformation, and then 5-FOA was used as a counterselecting agent to select against the *URA3* plasmid containing the wild-type *HHT1-HHF1* genes. All yeast strains (listed in *Supplementary file 1*) were derived from histone H3/H4 'shuffle' strain MSY421 from M. Smith and C.D. Allis (*Recht et al., 2006*), which has both chromosomal H3/H4-encoding loci deleted.

Genetic screens for asymmetric mutants were carried out in:
PKY4171
MATa; Δ(hht1-hhf1); Δ(hht2-hhf2); leu2-3,112; ura3-62; trp1; his3; bar1::hisG
+p(HHT1-HHF1, URA3, CEN)
DMS crosslinking studies:
PKY4171 derivatives with (HHT1-HHF1, URA3) plasmid replaced with:
PKY4625

+P22 (Myc-H3Y = Myc-hht2(L109I, A110W, L130I), HHF2, LEU2)
+P67 (bio-H3X-126A = biotin-hht2(L126A, L130V), HHF2, TRP1)
+P72 (V5-H3X-126A = V5-hht2(L126A, L130V), HHF2, URA3)

PKY4694
+P54 (biotin-H3Y = biotin-hht2(L109I, A110W, L130I), HHF2, LEU2)
+P72 (V5-126A = V5-hht2(L126A, L130V), HHF2, URA3)
+P35 (Myc-H3Y = Myc-hht2(L109I, A110W, L130I), HHF2, TRP1)

Mass spectrometry samples were grown in derivatives of
PKY4574 = PKY4171 with P65 (BirA::HIS3 integration plasmid) inserted.

## Directed evolution of improved H3X alleles

Plasmid P46 (hht2-(L126V, L130V), HHF2, TRP1) was used as the template to generate four PCR-generated libraries, each with randomized nucleotides at one of four H3 codons (109, 110, 126, 130). The full length PCR product was gel purified, self-ligated with T4 DNA ligase and transformed into competent E. coli Top10F' cells. 1 ml LB was added after heat shock transformation and the cells recovered for 1.5 hr at 37°C. 100 µl cells were plated to check the colony number, and the rest of the transformation was put into culture tube with another 2 ml LB and antibiotic to grow O/N before plasmid DNA miniprep (Zymo). The randomization of the desired codon in the four libraries of hht2 was confirmed by sequencing (*Figure 1—figure supplement 1B*).

The four libraries were each transformed separately into yeast strain PKY4171, which carries the plasmid-borne wild-type H3 (URA3). 153 transformants were picked for each library and patched onto both SD-Trp-Ura and FOA plates at the same time. A subset of isolates was found inviable after incubation at 30°C for 10 days on FOA, indicating inability to grow the absence of wild type H3. These isolates were therefore chosen for the second step. For each library separately, cells from each FOA-sensitive isolate were pooled into 25 ml of SD media, grown to an A600 of ~0.5 and transformed with plasmid P44 (H3Y, H4, LEU2) and plated on SD-Trp-Leu. These transformants were then replica-plated onto FOA media. Plasmids from resulting FOA-resistant colonies were analyzed by colony PCR and sequencing. Candidate plasmids were retested by retransformation into PKY4171 with or without P44. The plasmids that conferred robust growth on FOA media in the presence of P44 and no detectable growth in its absence encoded the hht2 variants termed 126A (carrying mutations 126A, 130V), 130A (126V, 130A), 130T (126V, 130T), 130Q (126V, 130Q).

## Effects of the 'pseudo-wild type' H3X/H3Y interface on yeast growth

As asymmetric nucleosomes cannot be made on a wild-type H3 backbone, all experiments by necessity involve comparisons to pseudo-wild type yeast carrying the H3X/H3Y pair. Pseudo-wild type yeast are viable, but grow slowly relative to the parental strain in YPD at 30°C (*Figure 4—figure supplement 1A*). In addition, they are temperature-sensitive and grow extremely poorly at 37°C. For this reason, temperature sensitivity in this system is assayed at 34°C (*Figure 4D–F*, *Figure 4—figure supplement 2*). Importantly, the temperature-sensitivity phenotypes observed at this temperature occur in the same sets of mutants that grow poorly in high salt or in the presence of DNA-damaging agents (*Figure 4*). Thus, although the X/Y histone interface provides a sensitized assay for some phenotypes, it nevertheless represents a useful and internally-consistent platform for exploring important and physiologically-relevant aspects of gene regulation.

## Dimethyl suberimidate-crosslinked nucleosomes

100 ml cell cultures were grown in synthetic dextrose media to maintain selection for the three histone-encoding plasmids, with 250 nM biotin added to favor biotinylation of the tagged H3. Cells were harvested in early log phase (A600 ~0.25), pelleted and transferred to 1.5 ml O-ring microcentrifuge tubes (Fisher #02-707-353). Cells were washed three times with 1 ml extraction (E) buffer (20 mM Na borate, pH 9.0, 0.35 M NaCl, 2 mM EDTA, plus 1 mM PMSF added freshly). Cells were resuspended in 900 µl E buffer, and 0.5 ml glass beads (0.5 mm, Biospec) were added to each tube. Cell walls were mechanically broken with three one-minute pulses in a Biospec beadbeater, with five-minute incubations on ice between pulses. The tube bottoms were punctured with a flame-

heated 26 gauge needle, placed into a 12 × 75 mm plastic tube and centrifuged for 2 min at 1500 rpm in a tabletop clinical centrifuge at 4°C. The liquid extracts were resuspended and transferred to 1.5 ml microfuge tubes. 1/10 vol freshly dissolved 11 mg/ml dimethyl suberimidate (Pierce) in E buffer was added, and samples were incubated with rotation at room temperature 90 min. For time point aliquots to be analyzed directly on gels, proteins were precipitated by addition of 1/10 vol 100% TCA. For material to be MNase-digested and immunoprecipitated, the crosslinker was quenched by addition of 50 mM Tris-Cl, pH 7.5 and further rotation for 15 min. 10 mM $MgCl_2$ and 1 mM $CaCl_2$ were added and tubes were equilibrated at 37°C for 5 min. 20 µl MNase (Worthington, 20 U/µl in 10 mM Tris-Cl pH 7.4) were added, tubes were inverted five times and incubated at 37°C, 20 min. Digestion was stopped by moving tubes to 4°C, adding 40 µl 0.25 M EDTA/EGTA and inversion to mix. Material was centrifuged at 8000 x g for 1 min, 4°C, and pre-cleared by incubation with CL2B-sepharose beads for 30 min 4°C and centrifugation at 8000 x g for 1 min, 4°C. The supernatant was the input for the pulldown assays. 30 µl of a 50% slurry of streptavidin-agarose beads was added to each tube, followed by rotation for 2 hr at 4°C. Samples were then centrifuged at 8000 x g for 1 min, 4°C and the beads were washed three times w/1 ml TE +2 M NaCl +0.2% Tween20, rotating at 4°C for 5 min each time. After the last wash all supernatant was removed and beads were resuspended in protein gel sample buffer. Proteins were separated on 17% SDS-PAGE gels and analyzed by immunoblotting; the 31kD H3-H3 crosslinked species was quantified on a BioRad Chemidoc system.

## Expression and purification of recombinant histones

Yeast heterodimer 'X' (L126A, L130V) and 'Y' mutations (L109I, A110W, L131I) were introduced into codon-optimized human histone H3 sequences lacking cysteines. This yielded human heterodimer 'X' with the following point mutations: C96S, C110A, L126A, and I130V; and human heterodimer 'Y': C96S, L109I, C110W (human histone H3 already has I131).

All wild-type and mutant human histone H3 proteins were purified according to the rapid purification protocol previously described (*Klinker et al., 2014*), with slight modifications. Specifically, codon optimised histone H3 (supplied by Entelechon – now Eurofins) was expressed from a pETM13 vector, transformed into the E. coli strain Rosetta (DE3) pLysS (Novagen). Bacteria were grown in LB medium (with 50 µg ml-1 kanamycin and 25 µg ml-1 chloramphenicol) at 37°C. Protein expression was induced with 0.25 mM IPTG (OD600 = 0.6) and cells were harvested after 4 hr by centrifugation. Cell pellets were resuspended in cold lysis buffer (7 M urea, 20 mM sodium acetate (pH 5.2), 200 mM NaCl, 1 mM EDTA and 5 mM β-mercaptoethanol) and cells lysed by sonication. The lysate was clarified by centrifugation at 75,600 g for 20 min and the supernatant filtered using a 0.45 µm syringe filter (HPF Millex, Millipore). The sample was injected onto connected HiTrap Q HP (5 ml) and HiTrap SP HP (5 ml) columns (GE Healthcare) pre-equilibrated in lysis buffer. Columns were washed with lysis buffer before removing the HiTrap Q column. Histone H3 was eluted from the HiTrap SP column with a 12 CV gradient into 60% elution buffer (lysis buffer with 1 M NaCl). Fractions containing histone H3 were pooled, concentrated and dialysed into 1 mM DTT. Finally, the histones were aliquoted and lyophilised for storage.

Human histone H4 (kindly provided by T.Bartke) and the H2A/H2B dimer were purified as previously described (*Miller et al., 2016*).

## Nucleosome reconstitution

Purified histones were used to attempt the assembly of histone octamers, as previously described (*Miller et al., 2016*). Briefly, histones dissolved in unfolding buffer (20 mM Tris (pH 7.5), 6 M guanidine hydrochloride, and 20 mM DTT) were mixed in equimolar ratios and diluted to a final concentration of 1 mg ml-1. The histones were extensively dialysed against refolding buffer (10 mM Tris (pH 7.5), 2 M NaCl, 1 mM EDTA and 5 mM β-mercaptoethanol) before being concentrated and purified by size exclusion chromatography (Superdex 200 Increase 3.2/300 column; GE Healthcare). Pooled fractions were concentrated and combined with 167 bp Widom DNA at an equilmolar concentration, with the aim of reconstituting nucleosomes by salt deposition (*Luger et al., 1999*) using buffers containing 20 mM Tris (pH 7.5), 5 mM β -mercaptoethanol, 1 mM EDTA, and the following NaCl concentrations: 2 M, 850 mM, 650 mM and 150 mM. Following dialysis, the samples were centrifuged at

20,000 g to remove any precipitate before nucleosome assembly was assessed by native PAGE (5% polyacrylamide, 3% glycerol, 0.5 X TAE; 100 V for ~1.5 hr at 4°C).

## Mass spectrometric analysis of biotinylated H3 from asymmetric nucleosomes

1000 ml cultures were grown in YPD +250 nM biotin at 30°C to mid-late log phase (A600 = 0.4–0.7). Then, sodium azide and phenylmethylsulfonyl fluoride were added to the medium at final concentrations of 0.1% and 5 mM, respectively. Cells were pelleted and collected into two to four O-ring tubes as above, and washed twice with buffer TG (0.1 M Tris-Cl, pH 8.0, 20% glycerol) + protease/deacetylase/phosphatase inhibitors: 1 mM PMSF, 0.5 µg/ml leupeptin, 0.7 µg/ml pepstatin, 1.0 µg/ml E64, 1.0 µg/ml aprotinin, 1 mM benzamidine) + 20 mM Na butyrate, 2 mM nicotinamide, 1 mM EDTA, 1 mM Na3VO4. Tubes were frozen in liquid nitrogen and stored at −80C. All subsequent steps were on ice or at 4°C.

Each tube of frozen cells was resuspended in 900 µl TG buffer with the inhibitors described above, plus 0.5 ml glass beads (0.5 mm, Biospec). Cell walls were broken by bead beating, and beads were removed as described above. Liquid extracts were transferred to 1.5 ml eppendorf tubes, and centrifuged 7 K rpm, 5 min (4.5K x g). The cytosolic supernatant was removed and the pellet was resuspended in 1 ml Buffer L (50 mM Hepes-KOH pH 7.5, 140 mM NaCl, 1 mM EDTA, 1% Triton X-100, 0.1% sodium deoxycholate)+inhibitors as described above. Chromatin was sheared in a Bioruptor (Diagenode) for 30 min (2 × 15 min rounds of 30'' on/30' off on 'High' power; samples were chilled on ice 10 min between rounds). Extracts were transferred to microfuge tubes, centrifuged at 6 K rpm (=3.3K x g) for 10 min. The supernatant was the input for the pulldown assays. 30 µl of a 50% slurry streptavidin-agarose beads was added to each tube, followed by rotation for 2 hr at 4°C. (Beads were pre-equilibrated and blocked in Buffer L + inhibitors + 10 µg insulin per sample.) Samples were centrifuged at 8000 x g for 1 min, and the beads were washed three times with 1 ml TE +2 M NaCl +2 M urea, rotating at 4°C for 5 min each time. To remove salt for mass spectrometry, beads were washed three times with water, snap-frozen and stored at −80°C.

## On-bead histone propionylation and tryptic digestion

Biotin-tagged histone H3 bound to streptavidin beads were washed twice with 300 µl of 100 mM ammonium bicarbonate. For in vitro propionylation of histone H3, 300 µl of a 3:1 isopropanol:propionic anhydride mixture was added followed by ammonium hydroxide to adjust pH to ~8. The beads were incubated at 51°C for 20 min. Beads were then collected by centrifugation at 1000 rpm for 60 s and washed with ammonium bicarbonate before a second round of propionic anhydride treatment. The washed beads were resuspended in 50 mM ammonium bicarbonate and incubated with 2 µg of Promega sequencing grade trypsin overnight at 37°C. The digest was then derivatized with propionic anhydride using the same protocol as described above. Histone peptides were extracted from the beads with 200 µl of 50% acetonitrile in water. Peptides were completely dried in a SpeedVac concentrator and then resuspended in 0.1% triflouroacetic acid for MS analysis.

## Liquid chromatography single reaction monitoring mass spectrometry (LC-SRM-MS)

Histone peptides were injected in triplicate onto a TSQ Quantum mass spectrometer (ThermoFisher Scientific) directly linked to a Dionex nano-LC system. Peptides were first loaded onto a trapping column (3 cm × 150 µm) and then separated with an analytical capillary column (10 cm × 75 µm). Both were packed with ProntoSIL C18-AQ, 3 µm, 200 Å resin (New Objective). The chromatography gradient was achieved by increasing percentage of buffer B from 0% to 35% at a flow rate of 0.30 µl/min over 45 min. Solvent A: 0.1% formic acid in water, and B: 0.1% formic acid in 95% acetonitrile. The peptides were then introduced into the MS by electrospray from an emitter with 10 µm tip (New Objective) as they were eluted from analytical column. The instrument settings were as follows: collision gas pressure of 1.5 mTorr; Q1 peak width of 0.7 (FWHM); cycle time of 3 s; skimmer offset of 10 V; electrospray voltage of 2.5 kV. Targeted analysis of unmodified and various modified histone peptides were performed.

Raw MS files were imported and analyzed in Skyline with Savitzky-Golay smoothing (MacLean et al., 2010). All Skyline peak area assignments were manually confirmed. Total peak

areas normalized to a non-modified H3 peptide were used to plot bar graphs representing relative proportions of distinct histone modifications. The relative abundances were determined from the mean of three technical replicates with error bars showing standard deviation.

## Affinity purification of salt-washed chromatin

These preparations were performed as described above for the mass spectrometry samples with the following modifications. 100 ml cultures were grown to late log phase ($OD_{600}$ = 1 to 3) in YPD + 250 nM biotin and cells were immediately subjected to bead beating. After removal of the cytosolic supernatant, pellets were washed three times with 1.0 ml Wash Buffer (10 mM Tris-Cl, pH 8.0, 1 mM EDTA, 0.1% Triton X-100, 0.5 M NaCl + protease inhibitors/PTM inhibitors as described above), spinning at 4.5K x g for 5 min at each step. The washed chromatin resuspended in 1.0 ml Wash Buffer and sheared in a Bioruptor as described above. The soluble chromatin was fractionated using streptavidin-agarose as described above, except that the washes were performed three times with 1.0 ml Wash Buffer. Bound material from each preparation was resuspended in 50 μl SDS-PAGE gel sample buffer.

## Northern blot analysis

Cells were grown in YPD medium to mid-exponential phase ($OD_{600}$ = 0.4 to 0.7), and collected by filtration. RNA was prepared by a hot phenol method (*Collart and Oliviero, 2001*). Northern blot analysis was performed as described previously (*Morohashi et al., 2006*) with following modifications: *STE11* and *FLO8* PCR fragments (*Pattenden et al., 2010*) were used as probes that were labeled with alkaline phosphatase using Amersham AlkPhos Direct Labeling and Detection Systems (GE healthcare). Blots were exposed to HyBlot CL Autoradiography Film (Denville Scientific). Signal intensity was quantified by using ImageJ.

Primers for Northern probe (*Pattenden et al., 2010*):
o-STE11-11: GAA GGA GTT ACA TCA TGA GAA CAT TGT TAC
o-STE11-12: GTG TGC ATC CAG CCA TGG ATG CTG CAG CAA
o-FLO8-13: GAC GCT CAG AAG CAA AGA AGT TCT AAG GTA
o-FLO8-14: CTC AAC ACG TGA CTT CAG CCT TCC CAA TTA

# Acknowledgements

We thank members of the Rando and Kaufman labs for discussions and critical reading of the manuscript. This work was supported by NIH grant R01GM100164 (OJR and PDK), ERC Grant 340712 (NF), and the ISF I-Core on Chromatin and RNA in Gene Regulation (NF). Proteomics experiments were supported by NIH grants P30CA060553 and P41GM108569. AA is grateful to the Azrieli foundation for the award of an Azrieli Fellowship.

# Additional information

## Funding

| Funder | Grant reference number | Author |
| --- | --- | --- |
| National Institute of General Medical Sciences | R01GM100164 | Yuichi Ichikawa<br>Caitlin F Connelly<br>Hsin-Jung Chou<br>Hsuiyi V Chen<br>Oliver J Rando<br>Paul D Kaufman |
| European Commission | 340712 | Alon Appleboim<br>Hadas Jacobi<br>Nir Friedman |
| National Institute of General Medical Sciences | P41GM108569 | Nebiyu A Abshiru<br>Yupeng Zheng<br>Paul M Thomas<br>Neil L Kelleher |

| National Institutes of Health | P30CA060553 | Nebiyu A Abshiru |
| | | Yupeng Zheng |
| | | Paul M Thomas |
| | | Neil L Kelleher |

The funders had no role in study design, data collection and interpretation, or the decision to submit the work for publication.

## Author contributions

Yuichi Ichikawa, Conceptualization, Data curation, Formal analysis, Investigation, Writing—review and editing; Caitlin F Connelly, Conceptualization, Formal analysis, Investigation; Alon Appleboim, Resources, Investigation, Writing—review and editing; Thomas CR Miller, Nebiyu A Abshiru, Yupeng Zheng, Investigation; Hadas Jacobi, Daniel NA Bolon, Formal analysis; Hsin-Jung Chou, Resources, Investigation; Yuanyuan Chen, Resources, Validation; Upasna Sharma, Vineeta Bajaj, Resources; Paul M Thomas, Neil L Kelleher, Formal analysis, Supervision; Hsuiyi V Chen, Resources, Validation, Investigation; Christoph W Müller, Supervision, Investigation; Nir Friedman, Data curation, Formal analysis, Supervision; Oliver J Rando, Conceptualization, Data curation, Supervision, Funding acquisition, Writing—original draft, Project administration, Writing—review and editing; Paul D Kaufman, Conceptualization, Data curation, Supervision, Funding acquisition, Investigation, Writing—original draft, Project administration, Writing—review and editing

## Author ORCIDs

Neil L Kelleher (iD) http://orcid.org/0000-0002-8815-3372
Nir Friedman (iD) http://orcid.org/0000-0002-9678-3550
Oliver J Rando (iD) https://orcid.org/0000-0003-1516-9397
Paul D Kaufman (iD) http://orcid.org/0000-0003-3089-313X

## Decision letter and Author response

Decision letter https://doi.org/10.7554/eLife.28836.018
Author response https://doi.org/10.7554/eLife.28836.019

# Additional files

## Supplementary files

• Supplementary table 1. Yeast strain list.
DOI: https://doi.org/10.7554/eLife.28836.016

• Transparent reporting form
DOI: https://doi.org/10.7554/eLife.28836.017

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
