## [Decision Letter]

[Editors’ note: this article was originally rejected after discussions between the reviewers, but the authors were invited to resubmit after an appeal against the decision.]

Thank you for submitting your work entitled "A synthetic biology approach to probing nucleosome symmetry" for consideration by *eLife*. Your article has been favorably evaluated by a Senior Editor and three reviewers, one of whom is a member of our Board of Reviewing Editors. The following individual involved in review of your submission has agreed to reveal their identity: Song Tan (Reviewer #2).

Our decision has been reached after consultation between the reviewers. Based on these discussions and the individual reviews below, we regret to inform you that your work will not be considered further for publication in *eLife*.

The reviewers all felt that the methods were clever and novel and the manuscript was discussed in depth by the reviewers and editors. There were significant concerns that the WT X/Y dimers had reduced growth and acetylation levels relative to real WT. Hence some of the experimental observations may be affected by the sickness of these strains. Another issue was that the amount of novel biological information was limited. These issues are described in some detail in the reviews below.

*Reviewer #1:*

This is a very clever paper from the Kaufman and Rando lab that approaches very interesting issues in basic chromatin biology. The forced heterodimers of H3 mutants and their creation is very innovative and creative. The utility of this approach was illustrated by analyzing H3K36 methylation and Ser10 phosphorylation.

While the approach is very creative and innovative the biological discovery in the manuscript seems rather incremental for publication in *eLife*.

*Reviewer #2:*

The authors have developed a system for studying nucleosomes containing asymmetrically positioned histone H3 mutations or modifications in vivo. Using structure-based design combined with genetic screening, they have designed two yeast histone H3 variants which form obligate heterodimers in vivo. The H3X and H3Y variants support growth when expressed together from low-copy number plasmids in yeast strains with the chromosomal H3 copies deleted, in contrast to when only H3X or H3Y variants are expressed. By positioning appropriate mutations on H3X or H3Y, the authors are able to show that a single copy of H3K36 per nucleosome is sufficient to silence cryptic internal promoter in vivo. They further show that the H3S10A mutation (which eliminates phosphorylation on H3S10) affects Gcn5 acetylation of H3K9 in cis in vivo.

The method described is novel and interesting. The impact on the chromatin biology field is tempered by the limitation that, as described, the method is only applicable to introducing asymmetry on histone H3 residues. Mutations on histone H2A, H2B or H4 apparently cannot be studied using the H3X/H3Y pair, nor can histone crosstalks between different histones (as the authors point out). Given this limitation, I suggest modifying the title to: "A synthetic biology approach to probing nucleosome symmetry of histone H3 residues".

One aspect that was not discussed by the authors was the possibility of perturbation of nucleosome structure by the replacement of wild-type H3 protein with the H3X/H3Y pair. Such perturbations presumably might be subtle, and might affect only a subset of chromatin modification or remodeling enzymes. I do not believe this possibility affects the interpretation of the results they presented (with one possible exception), but I do recommend the authors acknowledge this possibility explicitly to make the reader aware of this potential complication. The possible exception is the lower fraction of K9ac for pseudo-wild type H3X/H3Y compared to wild type H3/H3 (Figure 5). This significant reduction in K9ac should be noted and possible interpretations/explanations provided.

The authors state that "Consistent with prior reports, we found that the fraction of H3 molecules acetylated at K9 was lower in yeast strains carrying H3S10A symmetrically, […]" Please provide the references for these prior reports. This might be my ignorance, but while I am aware of papers describing the H3S10 phosphorylation/H3K14 acetylation crosstalk, I am not aware of papers for H3S10 phosphorylation/H3K9 acetylation crosstalk. Why did the authors select the former crosstalk for study here instead of the latter?

*Reviewer #3:*

In this manuscript, the authors develop a nice system to allow specific tests of whether histone modifications are required to be symmetric or not. Using this system, they go on to test to interesting aspects of histone H3 function – the requirement for H3K36 methylation to repress cryptic promoters and whether a mutation, S10A, affects adjacent acetylation in cis and/or trans. The paper is clearly written and the results will be of general interest to the transcription community. In addition, they system itself will likely be of interest to other labs.

1) Subsection “Design and validation of obligate heterodimers”, third paragraph: – The text states that the genetic tests demonstrate that H3X or H3Y alone do not homodimerize. Please use another word in place of "demonstrate" such as "suggests." A strong conclusion about dimerization cannot be made from solely from this genetic test as the 5FOA sensitivity could be due to some other reason than a failure to dimerize.

2) Figure 2: – Why is H3-H3 the only crosslinked form of H3 that is observed? What about H3-H4 or higher-order crosslinks?

3) Figure 2: – While these results support that heterodimerization is preferred over homodimerization, it is difficult to assess how well it occurs compared to dimerization of wild-type H3. This is particularly true because the level of crosslinking is low. Therefore, additional controls would be helpful. For example, how efficient is the H3-H3 interaction if all tagged copies of H3 are wild-type?

4) Figure 2: – At least two more replicates should be done.

5) The results in Figure 3 suggest that the XY pairs grow considerably slower than wild-type yeast (90 minutes for wild type versus >3 hours for the XY "wild-type" strain). Is this because of the X and Y mutations, or something about the strain configurations, such as the plasmids being used to express the histone genes? It would be very helpful to include a comparison to a strain with wild-type H3 but otherwise the same configuration of histone genes.

6) Figure 3: – It would be helpful to include a key in the figure that indicates what the left, middle, and right bars are for each trio even though that information is in the figure legend.

7) Subsection “A single H3K36 suffices to silence cryptic internal promoters”, first paragraph: – In addition to references Carrozza et al., 2001; Keogh et al., 2005; Joshi and Struhl, 2005 and Govind et al., 2010, the authors should cite Drouin et al., 2010, Plos Genetics, 6, e1001173.

8) Figure 3: – The authors are assuming that the H3K36 methylation pattern previously observed in P38V mutants will occur in their XY system. For at least a subset of strains, this should be verified by western analysis, to measure H3K36me3 and H3K36me2, comparing mutant combinations to wild-type XY pairs. The XY strains in which this should be done are wt/wt, P38V/P38V, K36Q/K36Q, and P38V/K36Q.

9) In otherwise wild-type yeast, does K36Q in both H3 genes confer temperature sensitive growth at 34°C? If not, why is growth defective in these XY strains? Furthermore, that would mean that the phenotypes observed are synthetic phenotypes, observed for H3K36Q only when combined with the unknown effect of the XY system. Some comment on this should be provided.

10) Kumar et al. (Nature Comm. 5, 3965 (2014)) showed that yeast set2 mutants and H3K36A mutants confer hypersensitivity to DNA damaging agents such as phleomycin. It would just take a set of simple spot tests to test the K36 methylation requirement for this additional phenotype.

---

## [Author Response]

[Editors’ note: the author responses to the first round of peer review follow.]

The reviewers all felt that the methods were clever and novel and the manuscript was discussed in depth by the reviewers and editors. There were significant concerns that the WT X/Y dimers had reduced growth and acetylation levels relative to real WT. Hence some of the experimental observations may be affected by the sickness of these strains. Another issue was that the amount of novel biological information was limited. These issues are described in some detail in the reviews below.Reviewer #1:This is a very clever paper from the Kaufman and Rando lab that approaches very interesting issues in basic chromatin biology. The forced heterodimers of H3 mutants and their creation is very innovative and creative. The utility of this approach was illustrated by analyzing H3K36 methylation and Ser10 phosphorylation.While the approach is very creative and innovative the biological discovery in the manuscript seems rather incremental for publication in eLife.

We thank the reviewer for the positive comments. We view the chief novelty of this work to be the generation of a unique toolset. This resource will be broadly useful to the chromatin community for the exploration of an unexplored level of genomic regulation.

Reviewer #2:[…] The method described is novel and interesting. The impact on the chromatin biology field is tempered by the limitation that, as described, the method is only applicable to introducing asymmetry on histone H3 residues. Mutations on histone H2A, H2B or H4 apparently cannot be studied using the H3X/H3Y pair, nor can histone crosstalks between different histones (as the authors point out). Given this limitation, I suggest modifying the title to: "A synthetic biology approach to probing nucleosome symmetry of histone H3 residues".

Although we agree with this basic point, we found the suggested title problematic as it suggests that we are analyzing the symmetry of H3 amino acids, which might lead a reader to think we are analyzing D vs L amino acids. In addition, one can imagine eventual engineering efforts to make separable H3-H4 interfaces that, together with this H3-H3 interface, would enable broader probes of nucleosome symmetry. Given this, unless the reviewer finds this to be completely unacceptable, we felt the extant title was better than any alternative we could come up with.

One aspect that was not discussed by the authors was the possibility of perturbation of nucleosome structure by the replacement of wild-type H3 protein with the H3X/H3Y pair. Such perturbations presumably might be subtle, and might affect only a subset of chromatin modification or remodeling enzymes. I do not believe this possibility affects the interpretation of the results they presented (with one possible exception), but I do recommend the authors acknowledge this possibility explicitly to make the reader aware of this potential complication.

Indeed, the pseudo-WT X/Y strains do not grow as robustly as strains with unaltered histones, although the growth rates in the original Figure 3 were measured robotically in suboptimal conditions – a poorly aerated 96-well format. We now present growth data for the XY system in well-aerated YPD cultures in the new Figure 4—figure supplement 1, confirming that the XY interface does cause a growth defect as expected from mutations affecting the H3/H3 interface. That noted, all experiments involve comparisons to the pseudo-WT strains – thus internally controlling for this unavoidable issue – and we also now include more explicit discussion of the possible effects of the X-Y interface at several places in the text, as requested.

The possible exception is the lower fraction of K9ac for pseudo-wild type H3X/H3Y compared to wild type H3/H3 (Figure 5). This significant reduction in K9ac should be noted and possible interpretations/explanations provided.The authors state that "Consistent with prior reports, we found that the fraction of H3 molecules acetylated at K9 was lower in yeast strains carrying H3S10A symmetrically, […]" Please provide the references for these prior reports. This might be my ignorance, but while I am aware of papers describing the H3S10 phosphorylation/H3K14 acetylation crosstalk, I am not aware of papers for H3S10 phosphorylation/H3K9 acetylation crosstalk. Why did the authors select the former crosstalk for study here instead of the latter?

The effect of S10A on K9ac was especially strong and clear in our mass spectroscopy data so we focused on it here. As for why we ignored K14ac, in contrast to published reports which are based mostly on antibody-mediated detection, we actually found minimal effects of S10A mutation on H3K14ac in our mass spec data. This was true for S10A mutants not just in the X/Y system, but also on the wild-type H3 background – in other words, we find that S10A mutants affect K9ac but we cannot replicate prior reports on K14ac. Whether this discrepancy is due to “strain background” or something like this we do not know, but we felt it was wiser to simply avoid bringing up K14ac since it is really peripheral to the manuscript. Our K9ac data are reproducible and robust, being observed in multiple MS experiments and confirmed independently by Western blotting.

Also, the reviewer is correct, there isn’t a prior report that specifically examined the effect S10A on K9ac; we had misquoted older literature where some ChIP experiments had used antibodies that would recognize both K9ac and K14ac. We have corrected this.

Reviewer #3:[…] 1) Subsection “Design and validation of obligate heterodimers”, third paragraph: The text states that the genetic tests demonstrate that H3X or H3Y alone do not homodimerize. Please use another word in place of "demonstrate" such as "suggests." A strong conclusion about dimerization cannot be made from solely from this genetic test as the 5FOA sensitivity could be due to some other reason than a failure to dimerize.

Changed as requested.

2) Figure 2: Why is H3-H3 the only crosslinked form of H3 that is observed? What about H3-H4 or higher-order crosslinks?

We have detected these species using anti-epitope tag antibodies, testing for crosslinks between the biotin-H3 molecules captured on streptavidin-agarose and either V5- or Myc-tagged H3 molecules in the same nucleosome. Biotin-H3 molecules crosslinked to H4 would not be recognized by the anti-epitope antibodies (which would only be present on another H3 besides the biotin-tagged H3). We now mention this in the figure legend.

3) Figure 2: While these results support that heterodimerization is preferred over homodimerization, it is difficult to assess how well it occurs compared to dimerization of wild-type H3. This is particularly true because the level of crosslinking is low. Therefore, additional controls would be helpful. For example, how efficient is the H3-H3 interaction if all tagged copies of H3 are wild-type?

This analysis is now presented in new Figure 2—figure supplement 1 – DMS crosslinking of wild-type and X/Y interfaces is nearly indistinguishable.

4) Figure 2: At least two more replicates should be done.

This was also requested by reviewer 2, and has been performed as requested.

5) The results in Figure 3 suggest that the XY pairs grow considerably slower than wild-type yeast (90 minutes for wild type versus >3 hours for the XY "wild-type" strain). Is this because of the X and Y mutations, or something about the strain configurations, such as the plasmids being used to express the histone genes? It would be very helpful to include a comparison to a strain with wild-type H3 but otherwise the same configuration of histone genes.

As we noted above, the data from the original Figure 3 were measured robotically in a 96-well format, so the cells were not grown under optimal conditions. We now report growth rates comparing strains with wild-type histones and asymmetric histones grown in rich, aerated, rotating liquid media (new Figure 4—figure supplement 1).

6) Figure 3: It would be helpful to include a key in the figure that indicates what the left, middle, and right bars are for each trio even though that information is in the figure legend.

Included as requested.

7) Subsection “A single H3K36 suffices to silence cryptic internal promoters”, first paragraph: In addition to references Carrozza et al., 2001; Keogh et al., 2005; Joshi and Struhl, 2005 and Govind et al., 2010, the authors should cite Drouin et al., 2010, Plos Genetics, 6, e1001173.

Included as requested.

8) Figure 3: The authors are assuming that the H3K36 methylation pattern previously observed in P38V mutants will occur in their XY system. For at least a subset of strains, this should be verified by western analysis, to measure H3K36me3 and H3K36me2, comparing mutant combinations to wild-type XY pairs. The XY strains in which this should be done are wt/wt, P38V/P38V, K36Q/K36Q, and P38V/K36Q.

Included as requested in new Figure 4—figure supplement 3.

9) In otherwise wild-type yeast, does K36Q in both H3 genes confer temperature sensitive growth at 34°C? If not, why is growth defective in these XY strains? Furthermore, that would mean that the phenotypes observed are synthetic phenotypes, observed for H3K36Q only when combined with the unknown effect of the XY system. Some comment on this should be provided.

K36Q does cause ts growth in strains with wild-type H3 C-termini, and this is detectable at 37C in this case. This was shown in the original Figure 3—figure supplement 2. In an asymmetric XY background, the effect of the K36Q on temperature sensitivity is observed at 34C. Therefore, growth at 34C proved to be a convenient assay that accurately predicted the cryptic transcription phenotype of asymmetric histone strains. Notably, the cryptic transcription phenotype is not synthetic, as it is observed in strains expressing H3K36Q-wtC. In the revised manuscript we emphasize that although the XY histone interface provides a sensitized assay for some phenotypes, it nevertheless represents a useful platform for exploring important and physiologically relevant aspects of gene regulation.

10) Kumar et al. (Nature Comm. 5, 3965 (2014)) showed that yeast set2 mutants and H3K36A mutants confer hypersensitivity to DNA damaging agents such as phleomycin. It would just take a set of simple spot tests to test the K36 methylation requirement for this additional phenotype.

As requested, we have examined growth on MMS and phleomycin and observed that K36Q double mutants, but not single mutants, display elevated sensitivity. We include these data in new Figure 4. Thus, the stress sensitivity reported for double K36Q mutants is consistent across four stress conditions – high salt, high temperature, MMS, and phleomycin – demonstrating the generality of this finding.